# Local crystallization inside the polymer electrolyte for lithium metal batteries observed by *operando* nanofocus WAXS

Fabian A. C. Apfelbeck [1], Gilles E. Wittmann[2], Morgan P. Le Dû [1], Lyuyang Cheng[1], Yuxin Liang[1], Yingying Yan[1], Anton Davydok [3], Christina Krywka[3] & Peter Müller-Buschbaum [1] ✉

The development of next-generation lithium-based batteries is accompanied by the intention to suppress the formation of dendritic lithium on the electrode, and is dominated by the picture that dendrites start to grow at the electrodes. Shifting from liquid to solid-state electrolytes, a high transference number is a quantity that promises the restraint of such parasitic side reactions. In this study, nanofocus X-ray wide-angle scattering is used to detect possible lithium-based crystallites in the polymer-based electrolyte. We perform *operando* scanning nanofocus wide-angle X-ray scattering on a composite gel-type polymer consisting of poly(vinylidene fluoride-co-hexafluoropropylene) and the single-ion conducting polymer poly((tri-fluoromethane) sulfonimide lithium styrene) in a lithium symmetric cell. We observe the occurrence and kinetics of lithium carbonate crystallites inside the electrolyte over a depth of 16 μm during three half-cycles. Furthermore, we prove the existence of lithium hydroxide crystallites near the lithium electrode and their absence in the bulk. Importantly, we identify the growth of pure metallic lithium inside the electrolyte as a sign of lithium dendrite growth happening inside the polymer-based electrolyte and not at the electrodes. Thus, nanofocus wide-angle X-ray scattering visualizes local structure changes such as dendrite formation inside the polymer-based electrolyte despite an unchanged electrochemical performance.

Lithium metal is regarded as the ideal negative electrode for lithium batteries as it exhibits the lowest electrochemical potential (−3.04 V vs. standard hydrogen electrode) and the highest specific capacity (3860 mAh g$^{-1}$)[1]. However, the combination of lithium metal with conventional, non-aqueous liquid electrolyte (e.g., carbonate-based) is accompanied by uneven dendritic Li growth on the surface of the negative electrode, which leads to rapid performance loss, and even the explosion of the cells is possible. Besides tuning the composition of the electrolyte through additives[2] or passivating the electrode's

surfaces[3], the use of (solid) polymer electrolytes is a strategy to bypass the aforementioned issues due to thermal, electrochemical, and mechanical stability[4–6]. Among the variety of polymer electrolytes, single-ion conducting polymers are an interesting group as they exhibit a transference number close to unity due to the tethering of the anion to the polymeric backbone. According to theoretical models[7], this property is accompanied by the suppression of dendritic structures on the lithium metal surface. However, the usage of single-ion conducting polymers is debatable as it is reported that the benefits

[1]TUM School of Natural Sciences, Department of Physics, Chair for Functional Materials, Technical University of Munich, Garching, Germany. [2]Heinz Maier-Leibnitz-Zentrum (MLZ), Technical University of Munich, Garching, Germany. [3]Helmholtz-Zentrum Hereon, Geesthacht, Germany. ✉e-mail: muellerb@ph.tum.de

come only into play at impractically high currents and temperatures[8]. Therefore, the major goal of this study is to test if a single-ion conducting polymer can indeed suppress lithium dendrite formation and, if not, to observe where the dendrites form, i.e., if the picture that dendrite growth starts at the electrode is correct. For that reason, we make use of synchrotron-based nanofocus wide-angle X-ray scattering (nWAXS), as the nanometer-sized X-ray beam allows for scanning locally only the polymer electrolyte in the electrode-near area, in an orientation of the cell that prevents getting background from the electrodes. Using nWAXS, a high spatial resolution of the crystalline structure can be provided[9] and possible structures that are hardly visible with real space microscopy techniques can be resolved. Hence, the formation of possible lithium crystallites can be detected at the electrodes or inside the polymer-based electrolyte. Furthermore, in combination with the *operando* set-up, the temporal resolution is also achieved, and thus, physical-chemical processes can be studied.

In this work, we perform *operando* scanning nWAXS at room temperature on a symmetric lithium cell with a polymer-based electrolyte (PVDF-HFP/PSTFSILi in a mixture of ethylene carbonate and propylene carbonate (EC/PC)), which is specially designed for synchrotron experiments to spatially and temporally resolve the crystalline structure of the electrolyte on a nanoscale. With such an approach, we are able to identify rare crystallization events in the early stages of cell operation. Besides detecting metallic lithium in the polymer electrolyte, we observe the unexpected local kinetics of lithium carbonate and lithium hydroxide inside the polymer during three half-cycles over a depth of approximately 16 μm.

## Results

A gel-type polymer electrolyte (PVDF-HFP/PSTFSILi (1:1 wt%) in EC/PC (1:1 v/v)) is prepared by a classical solution-based approach similar to previously reported procedures[10]. The crystallinity of the pristine polymer electrolyte is investigated with grazing incidence wide-angle X-ray scattering (GIWAXS)[11] in a custom-made argon-flow cell (Supplementary Fig. 1). Fig. 1a displays the radial integration (so-called pseudo XRD) of the 2D GIWAXS data. A relatively broad halo in the low q regime with a maximum at around $q ~ 1.42\,\text{Å}^{-1}$ can be recognized. Specifically, the peak and its shoulder are characteristic of PVDF-HFP[12,13], whereas PSTFSILi does not exhibit any prominent features. Additionally, a broad, low intensity peak is present at $q ~ 2.65\,\text{Å}^{-1}$. Moreover, any further pronounced crystalline peaks cannot be identified. To investigate the polymer electrolyte with the nanofocus X-ray beam, a special cell has been developed in which a stack of battery materials can be placed. Here, only the polymer electrolyte is penetrated by the nanofocus X-ray beam in contrast to conventional *operando* X-ray transmission experiments in which all layers are usually illuminated. A comparison of a typical scattering experiment and the approach used in this study is schematically illustrated in Supplementary Fig. 2a, b. Additionally, a construction drawing and a photograph of the respective *operando* nWAXS cell and a corresponding electrochemical pre-test are given in Supplementary Figs. 2c–e and 3. In this study, a Li metal|polymer electrolyte|Li metal configuration is placed into the cell and brought to the nanofocus endstation of beamline P03 at DESY Hamburg, Germany (Supplementary Fig. 4). Before the *operando* experiment, a vertical X-ray scan along the polymer with a microbeam covering 21 positions in y direction is performed (Fig. 1b). Rather broad peaks at around $q ~ 2.25\,\text{Å}^{-1}$ and $q ~ 2.47\,\text{Å}^{-1}$ can be noticed in the vicinity of the lithium electrodes; however, in the polymer bulk, these crystallites are absent. Moreover, these peaks are not visible in the static GIWAXS measurement of the pristine polymer electrolyte film. These reflexes can be assigned to lithium hydroxide (LiOH, mp-23856), which is known to be an essential part of the SEI layer and the native oxide layer of lithium metal[14]. The origin of LiOH might stem from $H_2O$ contamination; however, as the peaks are not present in the static measurement, a formation of LiOH

crystallites in the electrolyte due to the contact of the polymer layer with the lithium metal electrodes is more likely. Moreover, as it has been previously described[15,16], the partially reversible electrochemical reaction of LiH and $Li_2O$ to LiOH + Li can also serve as a lithium hydroxide source. In addition to that, a peak at $q ~ 1.55\,\text{Å}^{-1}$ is randomly distributed. This peak matches quite well with the first Bragg peak of lithium carbonate ($Li_2CO_3$, mp-556777); however, no further $Li_2CO_3$ peaks can be identified. To confirm the identity of these crystallites as lithium hydroxide and lithium carbonate, we further perform Fourier-transform infrared (FTIR) spectroscopy measurements (sketch and photograph of the setup in Supplementary Fig. 5) on the pristine polymer and a polymer that was in contact with a lithium metal chip for eight days. The two FTIR spectra are shown in Fig. 1c, and a clear difference between these two spectra can be recognized as the contacted polymer shows a noticeable increase of stretching vibrations of the -C = O bond at $v(C = O) = 1775\,\text{cm}^{-1}$, which can be attributed to a rise carbonate groups in the polymer[17]. Furthermore, an emerging peak at $3566\,\text{cm}^{-1}$ can be assigned to stretching vibrations of the LiO-H bond of LiOH * $H_2O$ and is an indication of the presence of LiOH[17]. The absence of any O-H bond vibrations in the pristine polymer film indicates that there is no $H_2O$ contamination initially. Overall, the FTIR test shows two things: 1) The transition of compounds from lithium into the electrolyte is completely pressure independent and directly happens after contacting the materials at room temperature, and 2) the peaks in the radial integrations in Fig. 1b are not X-ray induced and thus not a measurement artefact but indicate indeed crystalline compounds in the polymer reliably. Further identifiable peaks in the FTIR spectra are indicated and assigned in Fig. 1c correspondingly ($v(C-H) = 3020\,\text{cm}^{-1}$ and $2980\,\text{cm}^{-1}$, $\delta(C-H) = 1450\,\text{cm}^{-1}$ and $1480\,\text{cm}^{-1}$, and $v(S-O) = 1390\,\text{cm}^{-1}$)[18,19]. Furthermore, a photograph of the lithium-contacted polymer is given in Fig. 1d. The contacted area exhibits a discoloration of the polymer from white to black, which is clearly visible by eye and indicates the transition of compounds from the lithium metal surface into the polymer. This finding is in good agreement with the results by Shadike et al., who also showed a photograph of extracted SEI material that appeared black[20]. Next, *operando* nWAXS measurements are performed by recording nWAXS data and simultaneously letting the cell run electrochemically. An illustration of the experimental idea is given in Fig. 1e. A $350 \times 330\,\text{nm}^2$ (HxV) sized X-ray beam scans the electrolyte film in a grid mesh pattern during lithium plating and stripping. The number of measurement points and, thereby, the resolution of the grid mesh is limited by the duration of the corresponding electrochemical reaction.

The collected scattering pattern is an average of the whole depth of the polymer electrolyte, while the scanning provides the spatial resolution parallel and perpendicular to the electrode. Furthermore, the transmission for an X-ray beam with an energy of 12.62 keV for PVDF-HFP with a depth of 5 mm is approximately 6.8%, while it is around 57.5% for polystyrene. Thus, the absorption is predominantly determined by PVDF-HFP. To obtain sufficient statistics, a counting time of 20 s is chosen in every step of the scanning. Throughout the whole *operando* nWAXS experiment, the $Li_2CO_3$ peak at $q ~ 1.55\,\text{Å}^{-1}$ is emerging and disappearing; however, not for every position in the scanned area. For example, radial integrations of the 2D detector data of the first, second, and third half cycle are shown in Fig. 2a–c. For clarity, the five radial integrations of each half-cycle are shifted along the abscissa of the plot as otherwise they would overlap. Here, at position $x = 2$ in the second half-cycle, the peak is very prominent, whereas for the other positions, it is relatively weak or non-existent. A mapping of this peak intensity, together with the electrochemical data, is shown in Fig. 2d–f. Note that each position in the mapping is measured at a different time but at a constant current and, thus, voltage. Interestingly, by comparing the plating and stripping maps (Fig. 2d–f), kinetics of this peak intensity and thus, the crystallites can be noticed. In the first half-cycle, the crystallites are more present in the lower area

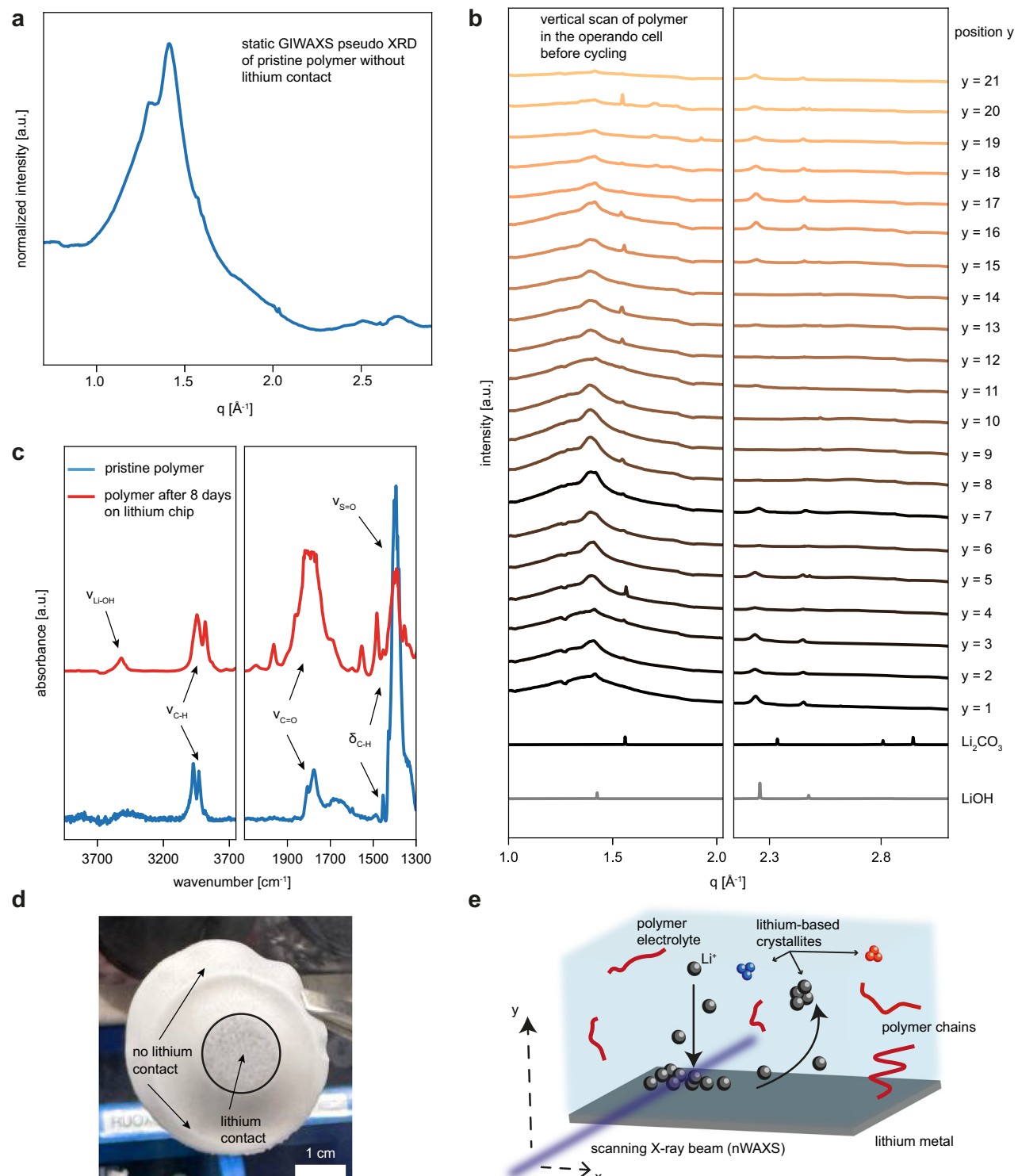

**Fig. 1 | Polymer electrolyte characterization and schematic of the *operando* scanning nWAXS experiment. a** Static GIWAXS radial integration (pseudo-XRD) of the pristine polymer film normalized to the photon flux. **b** Vertical nWAXS scan of the polymer before the *operando* experiment. **c** FTIR spectra of the pristine and lithium-contacted polymer film. **d** Photograph of the lithium-contacted polymer. **e** Illustration of the synchrotron experiment. The nano-sized X-ray beam penetrates and scans the polymer electrolyte during electrochemical cycling at room temperature. Source data are provided as a Source Data file.

of the polymer, whereas in the third half-cycle, these crystallites accumulate in the top region. This fact is of special interest, as half-cycles one and three have the same external voltage applied. In our opinion, the spatial redistribution can be caused by diffusion of the $CO_3^{2-}$ ions in the solvent phase, which can be accelerated by the applied external voltage and thus ends up in migration. Then, they can crystallize, dissolve, move, and re-crystallize. Hence, we can visualize

internal processes inside the polymer, which are not indicative of conventional electrochemical methods.

As expected, the overpotential of the *operando* cell is high. Besides room temperature, which results in lower ionic conductivity and a higher interfacial resistance, the major factor for this is the small pressure that is applied to the battery stack. In the electrochemical data, which is measured during the *operando* nWAXS experiment,

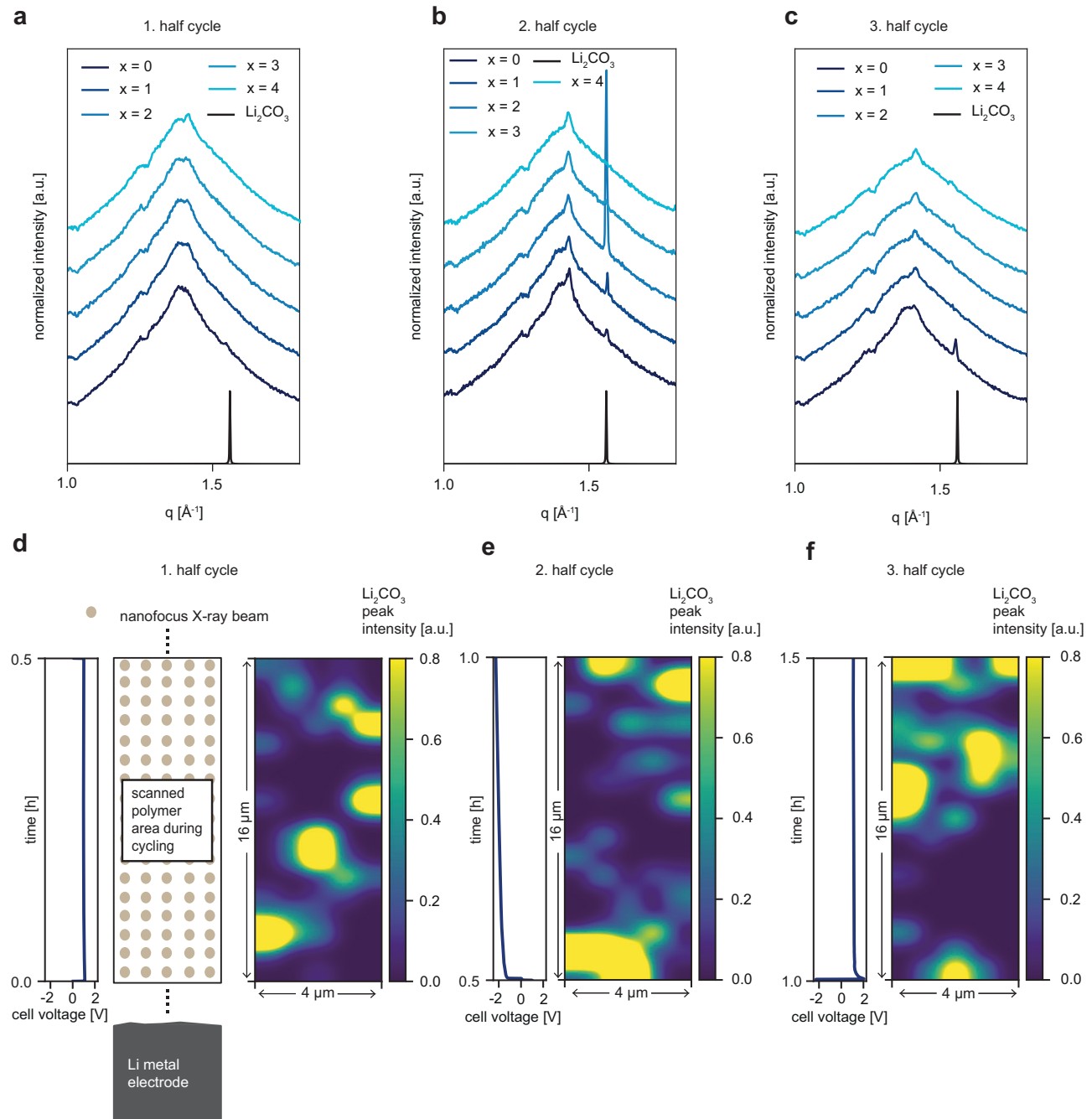

**Fig. 2 | *Operando* scanning nanofocus WAXS experiment. a–c** Selected radial integrations in the first (**a**), second (**b**), and third (**c**) half-cycle. The curves are shifted along the y-axis for clarity of the presentation normalized to the photon flux. **d–f** The polymer film is scanned at a sufficient height above the lithium metal electrode while lithium plating/stripping takes place, as shown in the schematic. 2D polymer maps of the Li₂CO₃ peak intensity combined with the electrochemical voltage profile during the first (**a**), second (**b**), and third (**c**) cycles. The 2D map shows the high kinetics of Li₂CO₃. The scan direction starts from the bottom and proceeds horizontally. The electrode position is indicated. Source data are provided as a Source Data file.

spikes, which could potentially indicate X-ray induced reduction, are not visible. Additionally, considering, for example, the bottom right corner of all three mappings, a possible radiation-induced crystallization[21] can be excluded, as no peaks appear in this region over the whole exposure time.

In addition to the Li₂CO₃ peak, the LiOH peaks at around $q \sim 2.25\,\text{Å}^{-1}$ and $q \sim 2.47\,\text{Å}^{-1}$ are permanently present in the scanned area, independent of the measured position in the electrolyte, as shown in Fig. 3a–c. Furthermore, higher orders of LiOH at $q \sim 3.34\,\text{Å}^{-1}$, $q \sim 3.47\,\text{Å}^{-1}$, and $q \sim 3.78\,\text{Å}^{-1}$ are identified. Lithium fluoride (LiF, mp-1009009) also has peaks theoretically located at $q \sim 2.47\,\text{Å}^{-1}$ and

$q \sim 3.47\,\text{Å}^{-1}$. However, the ratio of the experimental intensities for example at $q \sim 3.47\,\text{Å}^{-1}$ and $q \sim 3.78\,\text{Å}^{-1}$ matches very well with the theoretical ratio of the LiOH peaks ( ~ 1.17 vs. -1.13), which suggests that the most intense theoretical LiF peak at $q \sim 3.47\,\text{Å}^{-1}$ is not present and thus, also the peak at $q \sim 2.47\,\text{Å}^{-1}$ cannot be assigned to LiF. As described by Tan et al., the formation of LiF needs the presence of an NMC811 cathode material and a high voltage to produce LiF at least in the SEI[15]. Besides LiOH, the peak positions at $q \sim 3.34\,\text{Å}^{-1}$ and $q \sim 3.78\,\text{Å}^{-1}$ could also correspond to lithium oxide (Li₂O, mp-75589); however, the theoretical peak intensities do not match the experimentally observed ones, as the theoretical peak of Li₂O at $q \sim 3.34\,\text{Å}^{-1}$ is higher than that at

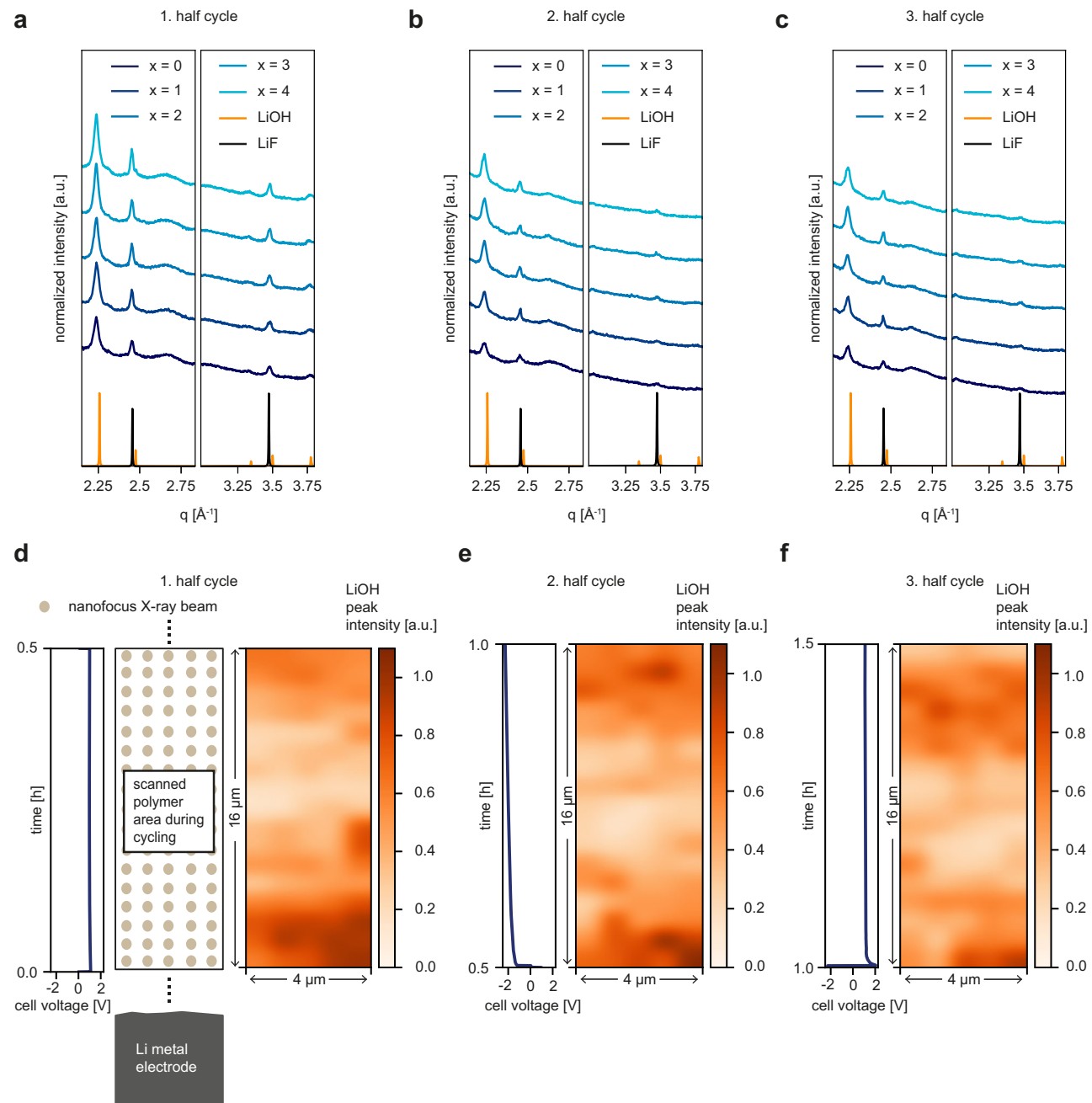

**Fig. 3 | *Operando* scanning nanofocus WAXS experiment. a–c** Selected radial integrations in the first (**a**), second (**b**) and third (**c**) half-cycle. The curves are shifted along the y-axis for clarity of the presentation normalized to the photon flux. **d–f** The polymer film is scanned at a sufficient height above the lithium metal electrode while lithium plating/stripping takes place, as shown in the schematic. 2D polymer maps of the LiOH peak intensity combined with the electrochemical data during the first (**a**), second (**b**) and third (**c**) cycles. The 2D maps show that the LiOH crystallites are present in the whole scanned area but hardly change over the three half-cycles. The scan direction starts from the bottom and proceeds horizontally. The electrode position is indicated. Source data are provided as a Source Data file.

$q \sim 3.78\,\text{Å}^{-1}$. The intensity of the LiOH peak at $q = 2.25\,\text{Å}^{-1}$ over the plating and stripping is visualized in a mapping (Fig. 3d–f). Interestingly, the intensity distribution of this peak in the selected area stays rather constant during electrochemical cycling. This finding suggests that the kinetics of these crystallites are quite limited, meaning that there is hardly any change in the LiOH crystallites over time. This finding can be understood as the fact that these crystallites do not move, dissolve, or recrystallize (= kinetics) in an extensive manner, and the overall polymer film preserves its overall position during the experiment and thus, possible variations in thickness can be neglected.

Besides the permanently present LiOH crystallite peaks, a reflex at higher q-values, specifically at $q \sim 2.52\,\text{Å}^{-1}$ appears at some positions.

This peak can be assigned to pure metallic lithium (Li, mp-51, Fig. 4a–c). A mapping of the intensity of this peak is given in Fig. 4d–f. In the first two half-cycles, these crystallites are barely present in the polymer and do not particularly distribute. However, in the third half-cycle, these crystallites start to accumulate, in particular in the upper right half of the scanned area. This incipient agglomeration is ascribed to a first indication of dendritic lithium growth. According to previous studies[22], dendrites typically originate from the electrode and are influenced by the applied pressure. In general, high pressures suppress dendritic formation. High pressure is not present in our cell as the screws are adjusted by hand. Therefore, any artificial suppression of lithium dendrites through cell pressure can be excluded. In addition to

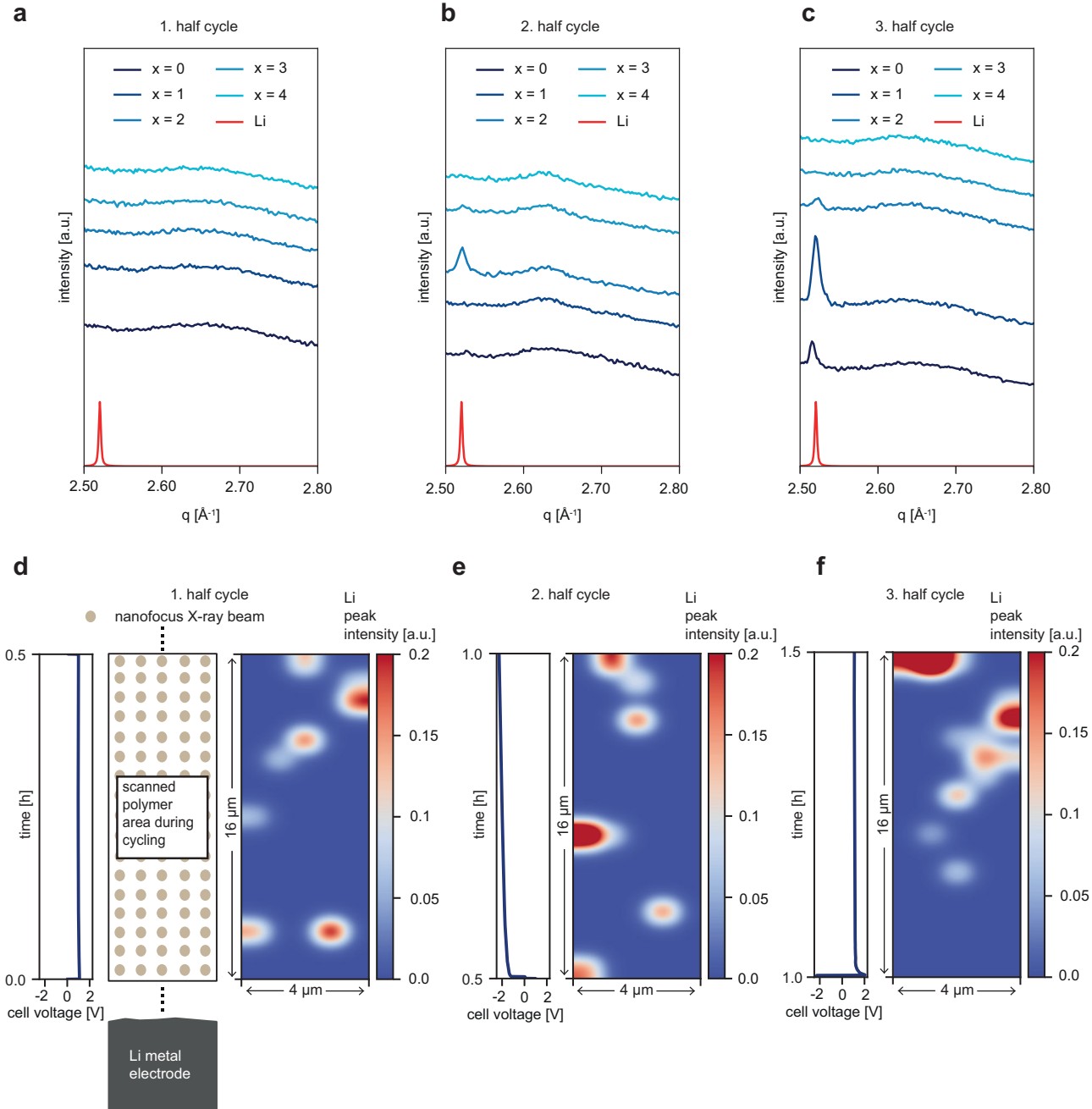

**Fig. 4 | *Operando* scanning nanofocus WAXS experiment. a–c** Selected radial integrations of the first (**a**), second (**b**) and third (**c**) half-cycle. The curves are shifted along the y-axis for clarity of the presentation normalized to the photon flux. **d–f** The polymer film is scanned at a sufficient height above the lithium metal electrode while lithium plating/stripping takes place, as shown in the schematic. 2D polymer maps of the Li peak intensity combined with the electrochemical data during the first (**a**), second (**b**) and third (**c**) cycle. The 2D maps show that the lithium crystallites are only little present in the polymer but change position over time. The scan direction starts from the bottom and proceeds horizontally. The electrode position is indicated. Source data are provided as a Source Data file.

the aforementioned peaks, there are also reflexes that cannot be clearly identified (Supplementary Fig. 6).

Next, we study the impact of the observed crystallites in the polymer electrolyte on its electrochemical properties. Therefore, electrochemical impedance spectroscopy measurements of this polymer electrolyte are performed with a stainless steel and lithium-symmetric cell over a time period of 96 h with 12 h steps. The corresponding Nyquist plots are depicted in Fig. 5a, b. The bulk resistance of the polymer is determined by the width of the (first) semicircle[23] and the extracted resistance values of both cells are compared in Fig. 5c. As expected, the trend of both resistance curves until equilibrium is the same; however, the bulk resistance from the lithium-symmetric cell is around 20 Ω higher at every time compared to the stainless-steel symmetric cell. Furthermore, a transference number test is conducted by performing a combination of chronoamperometry with electrochemical impedance spectroscopy before and after polarization[24]. The width of the second semicircle, which represents the interfacial resistance, is used to estimate the transference number to be -0.68 (Supplementary Fig. 7). Additionally, the electronic conductivity is determined by applying a voltage of 100 mV to a stainless-steel cell (blocking conditions) similar to Han et al.[25] A steady-state current of around $I = 0.8$ nA is set in after 1 h (Fig. 5d). This current is low but non-zero and thus non-negligible.

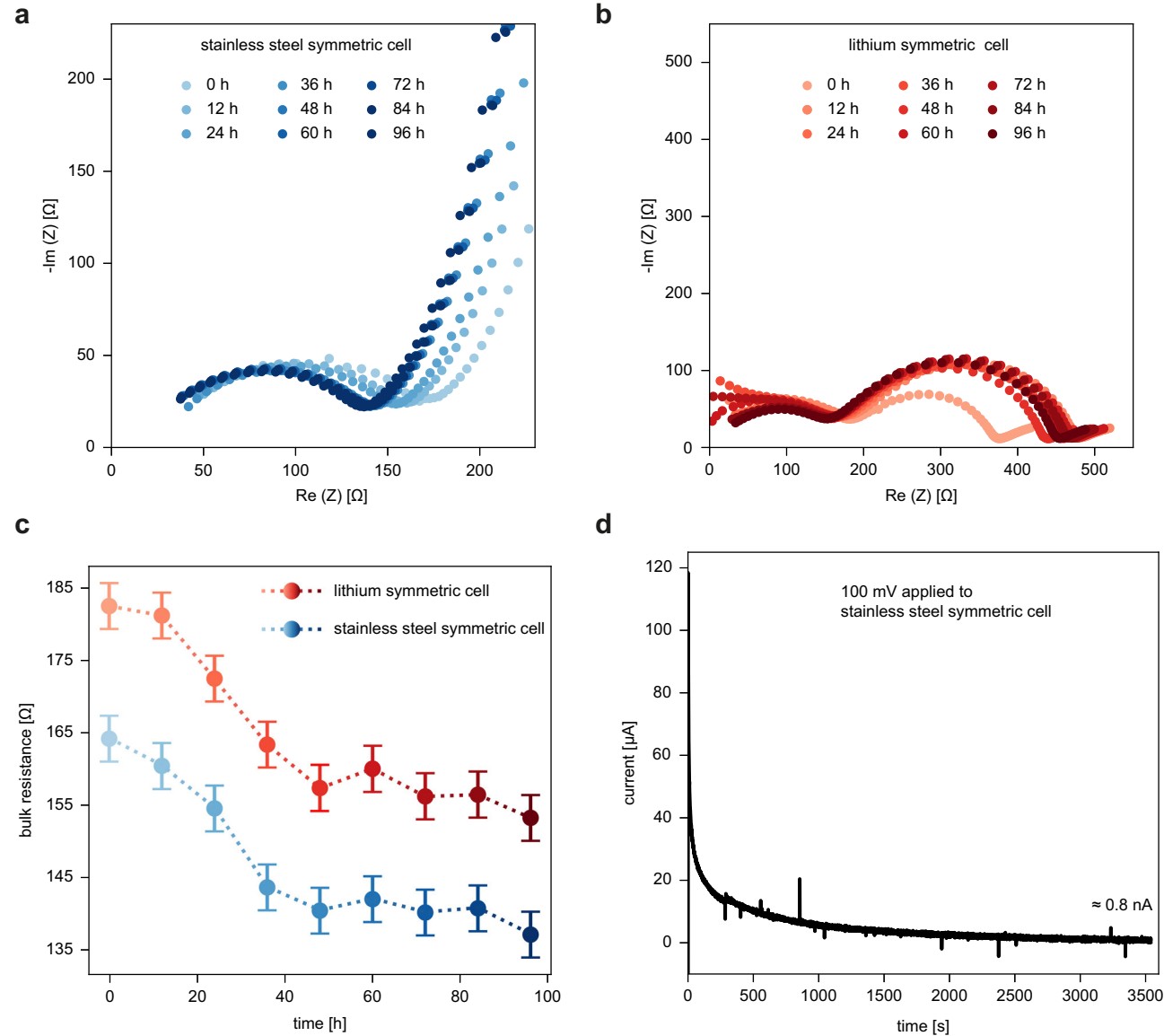

**Fig. 5 | Electrochemical characterization of the polymer electrolyte.** Nyquist plots of the polymer electrolyte in a stainless steel (**a**) and a lithium (**b**) symmetric cell over a time period of 96 h. **c** Ionic bulk resistances plotted over time for both cells. The uncertainty is estimated by the error of reading the value. **d** Electronic conductivity test. Source data are provided as a Source Data file.

## Discussion

In general, the observed crystallites inside the polymer-based electrolyte can be responsible for the higher ionic bulk resistance of the lithium symmetric cell compared to the stainless-steel cell, as they can act as an ion-conducting barrier (Fig. 5a−c). Therefore, the movement of the lithium ions can be hindered, especially as the ionic conductivity of bulk LiOH at room temperature is quite low[26]. $Li_2CO_3$ and LiOH are an essential part of the SEI layer and the native passivation layer of the lithium metal electrode surface. Consequently, considering the solubility of $Li_2CO_3$ and LiOH in EC and PC[27], which is rather high compared to other SEI layer compounds, a possible explanation for the presence of these compounds can be given: Due to the contact of the lithium electrode with the soaked electrolyte film, the crystallites of the lithium surface can be dissolved, forming $CO_3^{2-}$, $OH^-$, and $Li^+$, which diffuse into the electrolyte. This is validated by our FTIR measurements of the pristine and lithium-contacted polymer, which confirm the presence of $Li_2CO_3$ and LiOH compounds in a lithium-contacted polymer film. Then, the ions mainly move in the solvent phase, and one possibility is to nucleate and form crystallites in the solvent phase.

Another possibility is to accumulate and crystallize at the interfaces between the amorphous and crystalline regions of the polymer electrolyte, similar to grain boundaries in ceramic electrolytes or at defects. The nWAXS results suggest that the lithium carbonate crystallites either move and thus change their position or be (partially) dissolved and re-crystallize at a different position, in contrast to the LiOH crystallites, which tend to stay stable over time. Additionally, the presence of the anions $CO_3^{2-}$ and $OH^-$ can be responsible for lowering the total transference number. Therefore, it is of high importance to mitigate the dissolution of these compounds. This can be achieved, for example, by coating the lithium surface with a non-soluble layer with a sufficiently high ionic conductivity. Besides surface engineering, the general development of electrolytes and their additives should take the solubility of $Li_2CO_3$, LiOH, and other SEI compounds into account. Therefore, we propose to include corresponding experimental tests in their characterization. On top of that, we observe the early stages of the formation of pure metallic lithium inside the electrolyte, which is not necessarily connected to the lithium metal surface but can lead to dendritic structures. Metallic lithium formation in the bulk of ceramic

solid-state electrolytes has been observed by Han et al.[25] and very recently by Liu et al.[28] due to the reduction of lithium ions. Our study expands such observations to polycrystalline gel-type polymers, which typically contain PVDF-HFP and EC/PC. The formation of metallic lithium in the polymer bulk can be caused by a non-negligible electronic conductivity and the resulting reduction of Li⁺. This result is an important expansion of the current understanding of dendritic lithium growth in polymer electrolytes, which is thought to originate from the lithium metal surface[29]. To test the influence of the lithium crystallites on the long-term stability, we propose post-mortem ex-situ scanning nWAXS experiments as a future research direction. Several lithium (symmetric) cells with various electrolytes, surface coatings, or temperatures, and hence different lifetimes, can be investigated after the cells have been shortened. The extracted electrolytes can be investigated by scanning nWAXS in the same manner as in this study. Thus, possible correlations between lifetime and internal lithium crystallization can be explored. Especially higher temperatures can have an impact on the lithium crystallization process as reaction kinetics and solvent decomposition can be altered. The fact that also not identifiable reflexes show up during cell cycling, which are not initially present and only appear at certain spots rarely, underlines the complexity and uniqueness of the interplay between the electrolyte and the lithium metal and the corresponding SEI compounds. In addition to studying the electrolyte, depending on the materials transmission, our custom-made cell allows for investigating every single layer of the battery separately. Therefore, with our work using nano-sized X-ray beams, we open up the opportunity to further understand the components of lithium batteries and, in the next step, create better batteries.

## Methods

### Gel-type polymer electrolyte preparation

PVDF-HFP ($M_w$ ~ 400,000 g/mol, Sigma Aldrich) and PSTFSILi ($M_w$ ~ 231,722 g/mol, Specific Polymers) polymers were mixed in a 1:1 wt% ratio in an Argon-filled glovebox ($H_2O < 0.1$ ppm, $O_2 < 0.1$ ppm, MBraun). Then, N-methyl-2-pyrrolidon (NMP, 99.8%, Carl Roth) was added and stirred for 12 h until a homogeneous solution ($c = 100$ mg/mL) was obtained. Afterward, the solution was poured into a Teflon mold and dried in an oven for 24 h at $T = 80\,°C$. Finally, a free-standing film with a thickness of around 80 μm was obtained and cut into disks with a diameter of 20 mm. Finally, the film was placed into a solution of EC/PC (1:1 v/v, 99.995%, Guotai-Huarong Commercial New Material Co., Ltd.) for 12 h, and the fraction of liquid is ~ 30 wt% of the final film.

### Electrochemical characterization

Stainless steel (304 stainless steel, diameter ~16 mm, thickness ~500 μm) and Li-symmetric (Li 99.9%, diameter ~16 mm, thickness ~ 400 μm, used as received without further treatment and stored in Argon-filled glovebox) CR2032-type coin cells (all from TMAX Battery Equipments) with the previously fabricated gel-type polymer electrolyte were fabricated in an Argon-filled glovebox ($H_2O < 0.1$ ppm, $O_2 < 0.1$ ppm, MBraun, $T = 20\,°C \pm 2\,°C$, applied closing pressure on coin cell 1000 psi). Potentiostatic electrochemical impedance spectroscopy ($V_a = 10$ mV, $f$ = from 1 Mhz to 0.1 Hz, $N_d = 10$ points per decade, no quasi-stationary potential, EC-Lab) and chronoamperometry tests were performed with a BioLogic VMP300 potentiostat at $T = 20\,°C \pm 2\,°C$ at normal lab conditions (0.01 V applied to Li-symmetric cell for transference number test and 0.1 V applied to stainless steel symmetric cell for electronic conductivity test). The transference number was calculated with the following formula:

$$t = \frac{I_s(\Delta V - I_0 R_0)}{I_0(\Delta V - I_s R_s)}$$

$I_s$: steady state current, $I_0$: initial current, $\Delta V$: applied voltage, $R_0$: resistance before polarization, $R_s$: resistance after polarization.

### Grazing incidence wide-angle X-ray scattering (GIWAXS)

Static GIWAXS measurements of the free-standing polymer film were measured at the microfocus station of P03 beamline at DESY, Hamburg, Germany[30]. A beam energy of 11.88 keV ($\lambda = 1.044$ Å) and a sample-to-detector distance of 183 mm were used. To avoid any degradation due to moisture, the sample was placed into a custom-made argon-flow cell. 1D integrations were obtained with the software INSIGHT[31].

### FTIR measurements

The pristine and lithium-contacted polymers were placed into an air-tight FTIR sample holder in an argon-filled glovebox ($H_2O < 0.1$ ppm, $O_2 < 0.1$ ppm, MBraun), respectively, to avoid air exposure of the material. After that, the sample holder with the respective sample was directly measured at ambient conditions (but the polymer film was still under an Argon atmosphere), and the beam was aligned to the respective polymer area. The FTIR sample holder was custom-made from copper to operate FTIR spectroscopy. Since the presented FTIR spectra are performed in transmission mode, the cell is equipped with IR transmissive ZnS windows, which close the sample environment with the help of a sealing O-ring. The copper cell can host sample sizes up to $1 \times 1$ cm². A sketch and photograph of the cell are given in Supplementary Fig. 5. The FTIR spectra were recorded with a Bruker Equinox 55 spectrometer and normalized to the peak at 3020 cm⁻¹.

### *Operando* transmission cell

The custom-made circular *operando* transmission cell with a PVC-based body had a diameter of 2 cm and a depth of 5 mm. For more details, a construction drawing is given in Supplementary Fig. 3. To avoid moisture penetration into the cell, a 50 nm layer of titanium was sputtered on Kapton foils, which were used as X-ray windows. These foils were put on sealing rings and fixed between two stainless steel plates at the front and back with eight screws in total (four from both sides). The threads of the two screws from the top and bottom were wrapped with Teflon tape to protect the samples from moisture degradation, and the top part of the screws served as contacts for the potentiostat. The samples were cut in the corresponding dimensions (width 7.5 mm, depth 5 mm), and the lithium metal|polymer|lithium metal sandwich was placed in the cell. Additionally, flat copper chips were placed between the lithium metal and the screws to protect the soft lithium metal and prevent short circuits. Furthermore, they guarantee sufficient planeness of the lithium chip. The cells were fabricated and sealed in pouch bags in an Argon-filled glovebox ($H_2O < 0.1$ ppm, $O_2 < 0.1$ ppm, MBraun) and then transported from Munich to the beamline in Hamburg and measured after 6 days of fabrication.

### Nanofocus wide-angle X-ray scattering (nWAXS)

*Operando* scanning nWAXS experiments were performed at the nanofocus endstation of the P03 beamline at DESY Hamburg, Germany[32]. A $350 \times 330$ nm² (HxV) sized X-ray beam with an energy of $E = 12.62$ keV ($\lambda = 0.98244$ Å) was used to penetrate the polymer electrolyte. A rough vertical scan along the cell stack was performed to identify each layer. The nano beam was aligned vertically and horizontally for the polymer bulk (but not the Li|polymer interface) to obtain a clear transmission signal with sufficient distance from the electrodes. The sample-to-detector distance (SDD) was set to ~215 mm, and the scattered light was collected by a Dectris Eiger 9 M detector. The cells were mounted in a 3D-printed sample holder and cycled with a BioLogic SP150 potentiostat ($I = 0.001$ mA, ~2.6 μA/cm²). Every half cycle, a mesh grid was recorded with an exposure time of $t = 20$ s for each measurement point. The mesh grid size covered $4 \times 16$ μm² (HxV) with $5 \times 17$ points, pronouncing the vertical direction of the polymer electrolyte. Furthermore, a relatively small horizontally scanning direction (4 μm of total 7.5 mm) was chosen to keep any effects

induced by a possible residual horizontal tilt at a minimum. A step size of 1 μm was chosen. 1D radial integrations from the 2D detector images were obtained with the software DPDAK[33] and normalized to the photon flux. The data analysis was done with self-written Python scripts. A Gaussian interpolation was used to visualize the peak intensities in the 2D mappings. The theoretical XRD data were simulated with VESTA based on data from Materials Project[34].

## Data availability

The source data used in this study are available in the mediatum database under accession code https://mediatum.ub.tum.de/1795516 (https://doi.org/10.14459/2025mp1795516).

## Code availability

The code used for data management and analysis is available at https://mediatum.ub.tum.de/1795516.

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

## Acknowledgements

This work was supported by funding from the Deutsche Forschungsgemeinschaft (DFG, German Research Foundation) under Germany´s Excellence Strategy—EXC 2089/1—390776260 (e-conversion), TUM.solar in the context of the Bavarian Collaborative Research Project Solar Technologies Go Hybrid (SolTech), the Center for NanoScience (CeNS), and the International Research Training Group 2022 Alberta/Technical University of Munich International Graduate School for Environmentally Responsible Functional Hybrid Materials (ATUMS). Y.L., Y.Y., and L.C. acknowledge the financial support from the China Scholarship Council (CSC). Parts of this research have been carried out at P03 beamline of the light source PETRA III, DESY, a member of the Helmholtz Association (HGF), Germany. We thank Reinhold Funer for manufacturing the *operando* cells, Christian L. Weindl for designing the 3D-printed

sample holder, Senay Öztürk for sputtering titanium on Kapton, Matthias Schwartzkopf for help at the MiNaX station of beamline P03, and Professor Yonggao Xia and Professor Ya-Jun Cheng for the supply of battery test equipment.

## Author contributions

F.A.C.A and G.E.W. developed and tested the nWAXS cell. F.A.C.A and P.M.B. designed the idea of the nWAXS experiment, including proposal writing. F.A.C.A., L.C., Y.Y., and Y.L. performed the nWAXS experiment. F.A.C.A and M.P.L. conducted and analyzed the FTIR measurements. All additional experiments were performed by F.A.C.A. A.D., and C.K. provided support and resources at the beamline. F.A.C.A. analyzed the data and wrote the manuscript. P.M.B. supervised the project and provided resources and project administration.

## Funding

## Competing interests

The authors declare no competing interests.
