## [Transparent Peer Review file · Nature Communications]

Local crystallization inside the polymer electrolyte for lithium metal batteries observed by operando nanofocus WAXS

Corresponding Author: Professor Peter Müller-Buschbaum

Version 0:

Reviewer comments:

Reviewer #1

(Remarks to the Author)

The manuscript explores the behavior of crystallites (Li_2CO_3 , LiOH , and metallic Li) within a single-ion conducting polymer electrolyte during electrochemical cycling using operando scanning nano-focus wide-angle X-ray scattering (nWAXS). While the authors provide interesting insights into localized crystallization phenomena, there are several critical shortcomings in the current work that prevent its recommendation for publication in Nature Communications.

1. Lack of Performance Analysis for Single-Ion Conducting Polymers:

Although the study emphasizes the relevance of single-ion conducting polymer electrolytes, the manuscript does not include any direct or comprehensive evaluation of their electrochemical performance. Without this analysis, it is unclear how the observed crystallization processes impact the ionic conductivity, cycling stability, or overall effectiveness of single-ion conducting polymers in lithium metal batteries. The manuscript fails to connect these fundamental observations with practical implications, which is crucial for advancing the field.

2. Testing Conditions May Not Reflect Practical Behavior:

The custom mold cells used for operando testing introduce experimental conditions that differ significantly from real battery operation. Specifically: Temperature Effects: The experiments were conducted at room temperature, while real-world performance is often assessed at 60°C . Elevated temperatures can significantly accelerate reaction kinetics and solvent decomposition, potentially skewing the crystallization processes.

Pressure Effects: The mechanical pressure exerted by the mold cells may artificially suppress dendrite formation or alter electrolyte behavior, leading to results that do not accurately reflect the internal processes in practical polymer electrolytes. Therefore, the results obtained under these conditions may not be generalizable to actual battery systems, and the authors do not address or validate these discrepancies.

3. Lack of Constructive Forward-Looking Insights:

While the authors observe the formation of crystallites such as Li_2CO_3 and LiOH , they do not discuss their potential impact on battery performance, nor do they propose any concrete strategies or solutions to mitigate these issues. The conclusion, "the influence of these crystallites on the performance of polymer-based batteries should be studied in more detail in the future," is vague and lacks instructive guidance for further research. A study aiming for publication in Nature Communications should provide actionable insights or innovative strategies to address the observed challenges.

There are several points which remain unclear and are listed below:

1. The phrase "poly(vinylidene fluoride-co-hexafluoropropylene)-based (PVDF-HFP) single-ion conducting polymer" in the abstract contains a scientific inaccuracy. PVDF-HFP is commonly used as an ion-conducting material but does not exhibit single-ion conductivity. While PVDF-HFP itself is not a single-ion conducting polymer, the authors are likely referring to a composite system where PVDF-HFP is combined with poly((trifluoromethane)sulfonimide lithium styrene) (PSTFSILi), which is a single-ion conductor. Therefore, to avoid potential confusion, the description should emphasize that the single-ion conductivity arises from the inclusion of PSTFSILi in the PVDF-HFP matrix.

2. (Figure 2): The paper reports the kinetics of lithium carbonate (Li_2CO_3) crystallite formation, where the intensity of the Li_2CO_3 peak shows clear variations during plating and stripping cycles. How do the observed variations in Li_2CO_3 crystallite formation correlate with the electrochemical performance of the polymer electrolyte (e.g., voltage or current profile)? Could these changes be indicative of any degradation processes that may affect long-term battery stability?

3. (Lines 142–145, Figure 2): In the radial integrations shown in Figure 2, the Li_2CO_3 peak appears at a q -value around 1.55 \AA^{-1} . Given the complexity of peak assignment in the operando nWAXS experiment, how confident are the authors in the

identification of this peak as Li_2CO_3 , especially considering that no additional characteristic Li_2CO_3 peaks were identified? Could there be a possibility of interference from other phases or artifacts from the measurement process?

4. (Figure 2c, Lines 145–149): The maps in Figure 2c show the intensity distribution of the Li_2CO_3 peak. Could you elaborate on why the Li_2CO_3 crystallites accumulate in the top region of the polymer electrolyte during the third half-cycle, even though the external voltage remains unchanged between the first and third cycles? What mechanisms could drive this spatial redistribution of crystallites?

5. (Lines 171–174): The paper discusses the challenge of assigning diffraction peaks to LiOH and LiF due to their overlapping diffraction patterns. How did the authors validate the peak assignments to LiOH , and could additional techniques, such as X-ray absorption spectroscopy (XAS) or nuclear magnetic resonance (NMR), provide complementary information to resolve these ambiguities more definitively?

6. (Lines 119–121): The electrochemical testing of the polymer electrolyte was conducted at 60°C . How does this temperature affect the solubility and mobility of lithium salts within the electrolyte? Would performing the same experiments at room temperature (instead of 60°C) potentially yield different crystallization kinetics, and how might this influence the interpretation of the results?

While the manuscript presents valuable experimental techniques and observations regarding crystallization behavior in polymer electrolytes, it falls short in connecting these findings to the practical performance of single-ion conducting polymers. Furthermore, the testing conditions and lack of actionable insights significantly limit the study's relevance and impact. The authors need to:

Include electrochemical performance metrics for single-ion conducting polymers.

Clarify and validate how experimental conditions (temperature and pressure) affect the observed results.

Provide specific, constructive proposals for future research directions or strategies to mitigate crystallite formation.

In its current form, the manuscript is not recommended for publication in Nature Communications.

Reviewer #2

(Remarks to the Author)

Reviewer #3

(Remarks to the Author)

The manuscript delivered the observation of the metallic lithium and LiCO_3 crystal formation in the polymer electrolyte using the operando nano-focus WAXS at the early-stage cycling of the battery. A Li-symmetric CR2032-type coin cell equipped with PVDF-HFP and PSTFSILi single ion conductor was prepared and tested. The author highlights that the formation of metallic lithium inside the electrolyte might be correlated to the dendrite formation. According to the author, this technique has been rarely reported to be implemented on polymer lithium metal batteries. However, from the reviewer's point of view, the manuscript has not yet fulfilled the criteria to be published in a top journal, and is recommended to be submitted to a more specialized journal. The major issues are:

1. Lack of details and discussion about how the state-of-the-art nano-focus WAXS technique is implemented in the batteries cycling process, or in other words, what are the difficulties and specific innovations of this approach.
2. Limited diversity in characterization methods and experimental validation to support the reliability of the observations. The study relies heavily on a single or narrow set of techniques, with limited experimental repetitions.
3. The discussion primarily focuses on presenting results without diving into the probable underlying mechanisms of the local crystallization inside the electrolyte but not the interface.

Some other comments and questions are listed below.

1. Figure or Fig.? This should be uniform in the manuscript, and the figure number should be given clearly. For example, Line 168, which figure is being discussed? Line 189, which figure in supplementary?
2. Some typo errors: Line 148 Fig. 2c-f, Fig. 2c-e? Line 172, the word H_2O , H_2O ?
3. Regarding the discussion in Line 188-192 and Supplementary Figure 4, it seems the peak of $q \sim 2.47 \text{ \AA}^{-1}$ is closer to that of LiF . It seems it is possible to have two species simultaneously. Since there is a time difference during the scan as discussed by the author, is there any influence on the result? Any comment on this?
4. In Figure 2 and Figure 4, why only two half cycles of radial integrations are given?
5. How many cells have been tested? Is it a common phenomenon of the formation of metallic lithium and Li_2CO_3 crystals inside the polymer electrolyte in such a battery system?
6. If this work focuses on tackling the technical issues of applying the operando nano-focus WAXS on the polymer electrolyte battery cell, more details and discussion about the approach should be given. If the point is to give new insight for exploring the SEI and dendrite formation in polymer electrolyte batteries, more characterizations and experimental validations that support the observation should be done. Additionally, it is necessary to provide more discussions on the mechanism and the dynamics process of local metallic lithium formation. For example, what are the mainstream viewpoints on the lithium dendrite formation process in polymer batteries, and what is the impact of the observation in this work on these viewpoints? And when the dendrite forms, whether the failure of the battery could be observed.

Reviewer #4

(Remarks to the Author)

Pr Muller-Buschbaum and colleagues report nanofocus WAXS to trace spatially resolved occurrence and development of crystalline species at the interface between a cycling Li metal electrode and a polymer electrolyte. They identify Li_2CO_3 , LiOH and Li metal at various distances from the bulk metal surface.

The interesting point of this study is that the presence and temporal evolution of crystalline compounds such as the mentioned ones can (potentially) be traced as a function of distance from an electrode during cycling. However, the key weaknesses which really need to be addressed before further consideration are

A) The experimental setup is too poorly described as to really be able to understand what is measured.

B) The scholarly presentation is very poor, seriously hampers understanding, and reminds me of an early draft. I even wonder if all authors read the manuscript as this was not caught.

C) What is the scientific insight (beyond showing the methodological possibility of measuring those crystalline compounds in time and space)?

Here are specific points further substantiating A) and B)

1) The research question is very diffuse and not clearly discernable from the abstract and intro, and hence, the learning is equally diffuse (C above). I warmly recommend this book, which shows how this could be tackled <https://global.oup.com/academic/product/writing-science-9780199760244?cc=at&lang=en>. The abstract mentions as motivation for the work "... the practical usage can be debated", which leaves the reader wondering why nWAXS could help. What does "Surprisingly, we observe the local kinetics of lithium carbonate, lithium hydroxide, and metallic lithium crystallites in the polymer electrolyte in the vicinity of the lithium electrode during three half-cycles over a depth of approximately 16 μm measured from the electrode" mean? Wasn't this the purpose? What is kinetics of crystallites? The introduction is with ~1000 words way too long (double of what it should be), wanders around in a review, and doesn't come to the point either. In my view it could be as simple as saying: We want to measure crystalline compounds in the electrolytes in-situ, with ~micron resolution up to some 10 micron away from the electrode and show by means of a polymer electrolyte on Li metal that this is useful. I'd buy that.

2) The "special cell", which is of course pivotal to perform this study is not at all explained at any reasonable depth of detail. Fig. S1 is, if I understand right, the transmission cell, where all layers are penetrated. Different from the "special capillary cell" mentioned in l. 70 of the intro. Or was the cell stack parallel to the screw faces and hence beam, which come from top and bottom into the cell stack? If so, the in-plane dimension of the cell was very large (4 mm or so like the M4 screws?). Or is it 5 mm as suggested in l. 139? What is the purpose of the glass capillary seen in Fig. S2b? Why would it not be possible to heat this cell to 60 °C? That doesn't appear too difficult. But the central question around this setup is (if I understood the setup right as mentioned) that your beam goes parallel to Li metal through ~4-5 mm of the polymer with 0 to 16 micron distance. How do you get the Li metal to a planeness of ~ micron and this plane with this precision parallel to the beam? If the Li metal surface has height variations of only

3) Fig. 2: it is not at all clear what $x = 0 \dots 4$ mean. Also you mention that "curves are shifted along the y axis for clarity of the presentation", y-axis of the scanning area or the abscissa of the plot? At what y height above the Li metal if this is meant by the sketch in c? Fig 2cde: what does it mean "combined with EC data"? the map is measured after 0.5, 1, 1.5 h in c,d,e? or during this 1/2h each? L. 148: there is no Fig. 2f. At what point has Fig. 1b been measured?

4) If Li_2CO_3 , mp 556777 means the materials project entry, it should be written as Li_2CO_3 , mp-556777.

5) L. 186: where are these broad peaks at 22.25 and 2.47 \AA^{-1} seen? In Fig S4? But at what point has it been measured? No indication in the At what point has Fig. 1b been measured? No such peaks there.

6) What does "kinetics of these crystallites are quite limited" in l. 183 mean? What is kinetics of crystallites? They move spatially or grow/shrink over time?

7) L. 211: "The observation of lithium crystallites matches quite well with the previously mentioned appearance of isolated lithium metal inside the polymer electrolyte". Where was this mentioned?

8) Fig. S5: are these non-identified peaks only at certain spots, do they evolve over time? Are the present initially?

9) Discussion: "However, our result of assigning the peaks to LiOH can be substantiated by Tan et al., who describe the necessity of an NMC811 cathode material and the high voltage to produce LiF at least in the SEI". Which paper you mean? Shouldn't it be cited here? What does this sentence mean? NMC811 produces LiF at high V. But how is this connected with LiOH at Li metal? L. 236: "SEI layer elements" elements or compounds? L. 243: what do you mean with "complexity and uniqueness of the system"? uniqueness that the present cell maybe is not representative of a non-operando cell with the same components? Is it then useful?

Version 1:

Reviewer comments:

Reviewer #1

(Remarks to the Author)

The authors' thorough and constructive responses to the initial round of reviews. The revised manuscript has significantly improved in both scientific rigor and clarity. In particular, the authors have made commendable efforts to address the core

concerns raised in the initial review. They have effectively correlated the observed crystallization phenomena with electrochemical performance metrics, clarified how experimental conditions such as temperature and pressure influence their findings, corrected and refined material terminology and phase assignments, and proposed specific strategies and future directions to mitigate crystallite formation in polymer electrolytes. The inclusion of EIS, transference number, and electronic conductivity measurements now provides a clear link between the observed crystallites and their impact on ionic conductivity. This strengthens the practical relevance of the operando nWAXS results. The discussion section is now better structured, with more insightful mechanistic interpretations and clearer language.

Specific Comments

1. Electrochemical Performance Integration

The added impedance, transference number, and electronic conductivity measurements are well executed and convincingly support the negative impact of crystallite formation on transport properties. The comparison between symmetric stainless steel and Li-Li cells is especially helpful.

2. Clarification on Experimental Conditions

The authors provided a reasonable rationale for conducting measurements at room temperature and have included a forward-looking plan to explore high-temperature behavior. Similarly, the clarification that no significant pressure was applied during cell assembly is well received.

3. Terminology Precision

The revised abstract and main text now correctly distinguish between PVDF-HFP and PSTFSLi, accurately attributing single-ion conductivity to the latter. This correction is important and appreciated.

4. Phase Identification of Li_2CO_3 and LiOH

The FTIR data and updated high-q radial integrations provide compelling support for the presence of LiOH rather than LiF. While complementary XAS/NMR would further enhance confidence, the authors acknowledge this limitation and make a sound case using available data.

5. Mechanistic Insight

The revised discussion offers a more coherent and insightful interpretation of the spatial redistribution and persistence of different crystalline phases. The analogy to grain boundary reduction in ceramics is particularly interesting and expands the relevance of the findings.

6. Improved Presentation

The manuscript has been revised for clarity and conciseness, particularly in the introduction. The explanations of the scanning geometry, beam alignment, and sample flatness now resolve previous ambiguities.

In conclusion, the authors have fully addressed my concerns and substantially improved the manuscript. I now support publication in Nature Communications.

Reviewer #2

(Remarks to the Author)

Reviewer #3

(Remarks to the Author)

The authors have thoroughly addressed the concerns. The manuscript may be published in its current form.

Reviewer #4

(Remarks to the Author)

The authors did make efforts to address the reviewer comments. The initially poor scholarly presentation improved a lot, but should still be improved. Methods need clarifications.

- 1) Experimental setup: I still don't get it how this far-from-precision-engineered and machined cell would allow a flatness of Li metal surfaces of <1 micron over the size of ~5 mm. I do have extensive experience with precision engineering and know well the limits of certain precisions (flatness, parallelity, parallel movement). You really want to say that two M5 screws that are screwed from the two sides into a plastic block have a parallelity to allow for a 350 nm beam passing in parallel to a Li metal plate less than 4 micron for 5 mm? These screws are sealed with a teflon tape. First, this is already for sealing purposes not good practice as sealing a thread with Teflon tape is done only for tapered threads where the gap between the male and female thread closes in as it is screwed in and this way it progressively seals. Doing the same on a parallel thread is "Pfusch". Second, this way the threads are definitely not kept precisely axial (which a thread is not meant for anyway). The front faces of the screws are not planar with precision. At least this is what the SI Fig. 2c shows. Also flat Cu plates will not make the Li/polymer interface more flat than it initially was. So how did you get the Li having less than a few micron deviation from planarity over the few mm? also the Cu is not shown in Fig. S2b. How did you determine the zero position above Li? How did you align the in-plane direction of the cell parallel to the beam? Was the cell on a z, tip, tilt stage? Fig S2b is not helpful beyond a rough sketch. Why don't you make a standard construction drawing with real sizes of your components in the region of the cell stack, i.e. the screws, the Cu, Li, polymer? What you want to hide?
- 2) Does Fig 2 and the others where positions relative to Li metal are shown really mean, that the lowest point is < 1micron

above the Li? Or do you simply mean that these 16 micron height scan are somewhere inside the polymer, sufficiently above the Li? But the methods at l. 342 claim that "the mesh grid size covered 4 x 16 μm^2 (HxV) with 5 x 17 points, pronouncing the vertical direction of the polymer electrolyte with respect to the electrode". As discussed above, it appears impossible that you can have such a precision above the Li.

3) The use of the word "kinetics" is still questionable. Kinetics has a meaning in physics, i.e. the proportionality of a flux in response to a driving force. But here it is simply used for a temporal change. This is the wrong use of the word and is confusing.

4) The abstract mentions Li crystallites. Is it this what you mean?

5) Methods:

a. in the section EC characterization, it is not clear how transference number was measured with EIS, can you clarify.

b. FTIR. What sample holder? ATR or transmission? Remember the methods should be described in a way that the experiments can be judged and repeated.

Version 2:

Reviewer comments:

Reviewer #4

(Remarks to the Author)

The main concern previously arose from the text and Figs. 2-4 which suggested that the beam passed very closely to the Li/polymer interface. Now saying that the beam passed "somewhere" clarified this question.

Also the other comments were well responded to. I'm happy to support publication.

Reply to Reviewer #1 comments to manuscript NCOMMS-24-74934:

The manuscript explores the behavior of crystallites (Li_2CO_3 , LiOH , and metallic Li) within a single-ion conducting polymer electrolyte during electrochemical cycling using operando scanning nano-focus wide-angle X-ray scattering (nWAXS). While the authors provide interesting insights into localized crystallization phenomena, there are several critical shortcomings in the current work that prevent its recommendation for publication in Nature Communications.

Response: We thank the reviewer for this feedback on our work and the much-valued remarks, which helped us to further strengthen the manuscript.

Comment: 1. Lack of Performance Analysis for Single-Ion Conducting Polymers: Although the study emphasizes the relevance of single-ion conducting polymer electrolytes, the manuscript does not include any direct or comprehensive evaluation of their electrochemical performance. Without this analysis, it is unclear how the observed crystallization processes impact the ionic conductivity, cycling stability, or overall effectiveness of single-ion conducting polymers in lithium metal batteries. The manuscript fails to connect these fundamental observations with practical implications, which is crucial for advancing the field.

Response: We thank the reviewer for bringing up this point, and we have added an extensive electrochemical characterization. Therefore, we performed further electrochemical impedance spectroscopy measurements on two coin cells. The polymer electrolyte sandwiched between 1) symmetric stainless steel and 2) symmetric lithium cell. The idea behind this comparison is to see if the bulk ionic conductivity in the lithium symmetric cell is lower due to the presence of the crystallites, as we proposed in the first draft. For that, we measure an impedance spectrum every 12 hours for 4 days and finally compare the resistances obtained from the Nyquist plots of both cells. As expected, both cells show the same trend in bulk resistance. However, we find that the bulk resistance obtained from the lithium symmetric cell is higher at every time compared to the symmetric stainless-steel cell (while having the same thickness). With this experiment, we can prove that the diffusion of lithium to form crystallites in the polymer electrolyte has a negative impact on the ionic conductivity of the polymer electrolyte. Furthermore, we conduct a chronoamperometry test combined with electrochemical impedance spectroscopy to estimate the lithium-ion transference number, which is around 0.68. Moreover, we conduct an electronic conductivity test according to Han et al. by applying 100 mV to a stainless steel symmetric cell, which yields a current of 0.8 nA. We have added this information in the revision.

It reads on pages 13 and 14:

“Next, we study the impact of the observed crystallites in the polymer electrolyte on its electrochemical properties. Therefore, electrochemical impedance spectroscopy measurements of this polymer electrolyte are performed with a stainless steel and lithium-symmetric cell over a time period of 96 h with 12 h steps. The corresponding Nyquist plots are depicted in Fig. 5a,b. The bulk resistance of the polymer is determined by the width of the (first) semicircle²³ and the extracted resistance values of both cells are compared in Fig. 5c. As expected, the trend of both resistance curves until equilibrium is the same, however the bulk resistance from the lithium-symmetric cell is around 20 Ω higher at every time compared to the stainless-steel symmetric cell. Furthermore, a transference number test is conducted by performing a combination of chronoamperometry with electrochemical impedance spectroscopy before and after polarization.²⁴ The width of the second semicircle, which represents the interfacial resistance, is used to estimate the transference number to be ~ 0.68 .

Additionally, the electronic conductivity is determined by applying a voltage of 100 mV to a stainless-steel cell (blocking conditions) similar to Han et al.²⁵ A steady-state current of around $I = 0.8$ nA is set in after one hour. This current is low but non-zero and thus non-negligible.”

We added a new Figure (Fig. 5) with the corresponding results. It shows Fig. 5 on page 15:

Comment: 2. Testing Conditions May Not Reflect Practical Behavior:

The custom mold cells used for operando testing introduce experimental conditions that differ significantly from real battery operation. Specifically: Temperature Effects: The experiments

were conducted at room temperature, while real-world performance is often assessed at 60°C. Elevated temperatures can significantly accelerate reaction kinetics and solvent decomposition, potentially skewing the crystallization processes.

Response: We thank the reviewer for bringing up the point regarding the experimental conditions. We agree in general that this kind of polymer electrolyte is usually tested at elevated temperatures to enhance the ionic conductivity and improve interfacial contacts. Therefore, the overall long-term performance is better at $T = 60\text{ }^{\circ}\text{C}$ compared to room temperature, and thus, the tests in the literature are predominantly conducted at $T = 60\text{ }^{\circ}\text{C}$. However, we think that the polymers should be tested at room temperature as this imitates real-world applications, for example, in electric vehicles, and is therefore more realistic. To keep consistency with the nWAXS experiments, we performed all further experiments at room temperature. Also, in addition, we agree with the reviewer that higher temperatures can significantly accelerate reaction kinetics and solvent decomposition, and added this information to the main text accordingly.

It reads on page 17:

“To test the influence of the lithium crystallites on the long-term stability, we propose post-mortem ex-situ scanning nWAXS experiments as a future research direction. Several lithium (symmetric) cells with various electrolytes, surface coatings, or temperatures, and hence different lifetimes, can be investigated after the cells have been shortened. The extracted electrolytes can be investigated by scanning nWAXS in the same manner as in this study. Thus, possible correlations between lifetime and internal lithium crystallization can be explored. Especially higher temperatures can have an impact on the lithium crystallization process as reaction kinetics and solvent decomposition can be altered.”

Comment: Pressure Effects: The mechanical pressure exerted by the mold cells may artificially suppress dendrite formation or alter electrolyte behavior, leading to results that do not accurately reflect the internal processes in practical polymer electrolytes. Therefore, the results obtained under these conditions may not be generalizable to actual battery systems, and the authors do not address or validate these discrepancies.

Response: We thank the reviewer for bringing up this point regarding pressure effects. The pressure of the screws on the sample stack is set by hand very gentle to avoid the breakage of the soft lithium metal. According to the literature, dendrite formation is indeed influenced by the applied pressure and is suppressed by rather high pressures. We added this point to the main text.

It reads on page 12:

“According to previous studies²², dendrites typically originate from the electrode and are influenced by the applied pressure. In general, high pressures suppress dendritic formation. High pressure is not present in our cell as the screws are adjusted by hand. Therefore, any artificial suppression of lithium dendrites through cell pressure can be excluded.”

Furthermore, we can validate that the dissolution and transition of compounds from the lithium surface into the electrolyte is pressure independent as we see from our FTIR experiment (see answer to comment #3) and thus, our findings are generalizable. Furthermore, the comparison of the ionic bulk resistances suggests the presence of crystallites in the polymer film hindering the lithium-ion movement (see answer to comment #1).

It reads on pages 5 and 6:

“Overall, the FTIR test shows two things: 1) The transition of compounds from lithium into the electrolyte is completely pressure independent and directly happens after contacting the materials at room temperature and 2) the peaks in the radial integrations in Fig. 1b are not X-

ray induced and thus not a measurement artefact but indicate indeed crystalline compounds in the polymer reliably.”

Comment: 3. Lack of Constructive Forward-Looking Insights:

While the authors observe the formation of crystallites such as Li_2CO_3 and LiOH , they do not discuss their potential impact on battery performance, nor do they propose any concrete strategies or solutions to mitigate these issues. The conclusion, “the influence of these crystallites on the performance of polymer-based batteries should be studied in more detail in the future,” is vague and lacks instructive guidance for further research. A study aiming for publication in Nature Communications should provide actionable insights or innovative strategies to address the observed challenges.

Response: We thank the reviewer for bringing up this point. As described above in our reply to comment #1, we further performed electrochemical tests, which show that the crystallites have a negative impact on the ionic conductivity of the polymer electrolyte. With these experiments, we studied the influence of the crystallites on the ionic conductivity and hence the performance of polymer-based batteries in more detail. Therefore, we have deleted the mentioned sentence and prove the influence through the corresponding data as shown in our reply to comment #1.

As we showed that the formation of crystallites is due to the dissolving of the surface layer, we strongly recommend surface engineering in combination with standardized solubility tests. We added this information to the main text.

It reads on pages 16 and 17:

“Therefore, it is of high importance to mitigate the dissolution of these compounds. This can be achieved, for example, by coating the lithium surface with a non-soluble layer with a sufficiently high ionic conductivity. Besides surface engineering, the general development of electrolytes and their additives should take the solubility of Li_2CO_3 , LiOH , and other SEI compounds into account. Therefore, we propose to include corresponding experimental tests in their characterization.”

Furthermore, based on our findings, we propose to include solubility tests in the development of electrolytes in their characterization.

It reads on page 17:

“To test the influence of the lithium crystallites on the long-term stability, we propose post-mortem ex-situ scanning nWAXS experiments as a future research direction. Several lithium (symmetric) cells with various electrolytes, surface coatings, or temperatures, and hence different lifetimes, can be investigated after the cells have been shortened. The extracted electrolytes can be investigated by scanning nWAXS in the same manner as in this study. Thus, possible correlations between lifetime and internal lithium crystallization can be explored. Especially higher temperatures can have an impact on the lithium crystallization process as reaction kinetics and solvent decomposition can be altered.”

Comment: There are several points which remain unclear and are listed below:

1. The phrase "poly(vinylidene fluoride-co-hexafluoropropylene)-based (PVDF-HFP) single-ion conducting polymer" in the abstract contains a scientific inaccuracy. PVDF-HFP is commonly used as an ion-conducting material but does not exhibit single-ion conductivity. While PVDF-HFP itself is not a single-ion conducting polymer, the authors are likely referring to a composite system where PVDF-HFP is combined with poly((trifluoromethane)sulfonimide lithium styrene) (PSTFSILi), which is a single-ion

conductor. Therefore, to avoid potential confusion, the description should emphasize that the single-ion conductivity arises from the inclusion of PSTFSILi in the PVDF-HFP matrix.

Response: We thank the reviewer for this very good comment, which helps us to enhance the precision of the manuscript. We have corrected the wording.

It reads on page 1:

“In this study, we perform pioneering operando scanning nano wide-angle X-ray scattering on a composite gel-type polymer consisting of poly(vinylidene fluoride-co-hexafluoropropylene) (PVDF-HFP) and the single-ion conducting polymer poly((trifluoromethane) sulfonimide lithium styrene) (PSTFSILi).”

Comment: 2. (Figure 2): The paper reports the kinetics of lithium carbonate (Li_2CO_3) crystallite formation, where the intensity of the Li_2CO_3 peak shows clear variations during plating and stripping cycles. How do the observed variations in Li_2CO_3 crystallite formation correlate with the electrochemical performance of the polymer electrolyte (e.g., voltage or current profile)? Could these changes be indicative of any degradation processes that may affect long-term battery stability?

Response: We thank the reviewer for raising this question. As mentioned above (reply to comment #1), we showed that the crystallites in the polymer electrolyte enhance the ionic bulk resistance. Therefore, the dissolution of SEI compounds generally is a process that worsens the performance of a lithium metal battery just by contacting the lithium electrode with the electrolyte (see FTIR experiment).

The power of the presented nWAXS experiment is that, electrochemically, there are no signs of any degradation processes as the voltage remains stable over the whole experiment. However, we make it possible to observe the internal crystallization inside the polymer electrolyte, which is not visible by means of conventional electrochemical techniques such as long-term cycling of lithium symmetric coin cells. We clarify this point in the revision.

It reads on page 8:

“Hence, we can visualize internal processes inside the polymer, which are not indicative of conventional electrochemical methods.”

The question to which extent the variations of the Li_2CO_3 crystallite can be indicative of any long-term processes cannot be answered definitively at this stage of the study, and thus we could only speculate about any effects. Nevertheless, we want to suggest future research directions to explore these correlations:

For making statements about the influence of the crystallites on the long-term stability, we propose post-mortem *ex-situ* scanning nWAXS experiments as a future research direction: Several lithium (symmetric) cells with various electrolytes or surface coatings and hence different lifetimes can be investigated after the cells have been shortened. The extracted electrolytes can be investigated by scanning nWAXS in the same manner as in this study. Thus, possible correlations between lifetime and internal crystallization can be found. We have added information to clarify this point.

It reads on page 17:

“To test the influence of the lithium crystallites on the long-term stability, we propose post-mortem ex-situ scanning nWAXS experiments as a future research direction. Several lithium (symmetric) cells with various electrolytes, surface coatings, or temperatures, and hence different lifetimes, can be investigated after the cells have been shortened. The extracted electrolytes can be investigated by scanning nWAXS in the same manner as in this study. Thus, possible correlations between lifetime and internal lithium crystallization can be

explored. Especially higher temperatures can have an impact on the lithium crystallization process as reaction kinetics and solvent decomposition can be altered.”

Comment: 3. (Lines 142–145, Figure 2): In the radial integrations shown in Figure 2, the Li_2CO_3 peak appears at a q -value around 1.55 \AA^{-1} . Given the complexity of peak assignment in the operando nWAXS experiment, how confident are the authors in the identification of this peak as Li_2CO_3 , especially considering that no additional characteristic Li_2CO_3 peaks were identified? Could there be a possibility of interference from other phases or artifacts from the measurement process?

Response: We thank the reviewer for this comment, which helps us to confirm the peak assignment of lithium carbonate. We can definitively exclude any artefacts from the measurement process as we further performed X-ray independent FTIR and electrochemical measurements. FTIR measurements were conducted on the pristine polymer electrolyte and a polymer electrolyte film, which was placed directly after the electrolyte soaking step onto a lithium chip and was then in contact for eight days. The corresponding FTIR spectra of both polymers show clear differences, as additional peaks appeared in the FTIR spectrum of the polymer that was in contact with lithium. We can identify LiO-H vibrations, which can be attributed to LiOH. Furthermore, we see a clear increase in C=O bonds in the polymer that was contacted with lithium. This is a strong indication of carbonate groups in the polymer, which further underlines our statement that there is material dissolving from the lithium surface and transitions from the lithium side into the electrolyte. Furthermore, we took a photograph of the polymer electrolyte after eight days of contact, and a clear change in the surface of the polymer is visible by eye. The area that was in contact with the lithium chip appears to be black. This observation is in accordance with the paper by Shadike et al., who also showed a photograph of the SEI material that appeared black.

It reads on pages 5 and 6:

“To confirm the identity of these crystallites as lithium hydroxide and lithium carbonate, we further perform Fourier-transform infrared (FTIR) spectroscopy measurements on the pristine polymer and a polymer that was in contact with a lithium metal chip for eight days. The two FTIR spectra are shown in Fig. 1c, and a clear difference between these two spectra can be recognized as the contacted polymer shows a noticeable increase of stretching vibrations of the -C=O bond at $\nu(\text{C}=\text{O}) = 1775 \text{ cm}^{-1}$, which can be attributed to a rise in carbonate groups in the polymer.¹⁷ Furthermore, an emerging peak at 3566 cm^{-1} can be assigned to stretching vibrations of the LiO-H bond of $\text{LiOH} \cdot \text{H}_2\text{O}$ and is an indication of the presence of LiOH.¹⁷ The absence of any O-H bond vibrations in the pristine polymer film indicates that there is no H_2O contamination initially. Overall, the FTIR test shows two things: 1) The transition of compounds from lithium into the electrolyte is completely pressure independent and directly happens after contacting the materials at room temperature and 2) the peaks in the radial integrations in Fig. 1b are not X-ray induced and thus not a measurement artefact but indicate indeed crystalline compounds in the polymer reliably. Further identifiable peaks in the FTIR spectra are indicated and assigned in Fig. 1c correspondingly ($\nu(\text{C-H}) = 3020 \text{ cm}^{-1}$ and 2980 cm^{-1} , $\delta(\text{C-H}) = 1450 \text{ cm}^{-1}$ and 1480 cm^{-1} , and $\nu(\text{S-O}) = 1390 \text{ cm}^{-1}$).^{18,19} Furthermore, a photograph of the lithium-contacted polymer is given in Fig. 1d. The contacted area exhibits a discoloration of the polymer from white to black, which is clearly visible by eye and indicates the transition of compounds from the lithium metal surface into the polymer. This finding is in good agreement with the results by Shadike et al., who also showed a photograph of extracted SEI material that appeared black.²⁰”

It shows Fig. 1 on page 7:

Comment: 4. (Figure 2c, Lines 145–149): The maps in Figure 2c show the intensity distribution of the Li_2CO_3 peak. Could you elaborate on why the Li_2CO_3 crystallites accumulate in the top region of the polymer electrolyte during the third half-cycle, even

though the external voltage remains unchanged between the first and third cycles? What mechanisms could drive this spatial redistribution of crystallites?

Response: We thank the reviewer for raising this question. In our opinion, the spatial redistribution can be caused by diffusion of the CO_3^- ions in the solvent phase, which can be accelerated by the applied external voltage and thus ends up in migration. Then, they can crystallize, dissolve, move, and re-crystallize.

It reads on page 8:

“In our opinion, the spatial redistribution can be caused by diffusion of the CO_3^- ions in the solvent phase, which can be accelerated by the applied external voltage and thus ends up in migration. Then, they can crystallize, dissolve, move, and re-crystallize.”

Comment: 5. (Lines 171–174): The paper discusses the challenge of assigning diffraction peaks to LiOH and LiF due to their overlapping diffraction patterns. How did the authors validate the peak assignments to LiOH, and could additional techniques, such as X-ray absorption spectroscopy (XAS) or nuclear magnetic resonance (NMR), provide complementary information to resolve these ambiguities more definitively?

Response: We thank the reviewer for this very helpful remark, which significantly improves the understanding of the results. Unfortunately, we are not able to perform XAS or NMR measurements. However, we added the radial integrations with the corresponding theoretical peak positions of LiOH and LiF at high q-values. For clarity, we deleted Figure 4 in the Supplementary Information and added the radial integrations of this q-range and the corresponding theoretical reflexes of LiOH and LiF to Fig. 3a, b, c. This representation is more comprehensive and indicates that the peaks are clearly identified as LiOH.

It reads on page 10:

“Furthermore, higher orders of LiOH at $q \sim 3.34 \text{ \AA}^{-1}$, $q \sim 3.47 \text{ \AA}^{-1}$, and $q \sim 3.78 \text{ \AA}^{-1}$ are identified. Lithium fluoride (LiF, mp-1009009) also has peaks theoretically located at $q \sim 2.47 \text{ \AA}^{-1}$ and $q \sim 3.47 \text{ \AA}^{-1}$. However, the ratio of the experimental intensities for example at $q \sim 3.47 \text{ \AA}^{-1}$ and $q \sim 3.78 \text{ \AA}^{-1}$ matches very well with the theoretical ratio of the LiOH peaks (~ 1.17 vs. ~ 1.13), which suggests that the most intense theoretical LiF peak at $q \sim 3.47 \text{ \AA}^{-1}$ is not present and thus, also the peak at $q \sim 2.47 \text{ \AA}^{-1}$ cannot be assigned to LiF. As described by Tan et al., the formation of LiF needs the presence of an NMC811 cathode material and a high voltage to produce LiF at least in the SEI.¹⁵“

It shows Fig. 3 on page 11:

Comment: 6. (Lines 119–121): The electrochemical testing of the polymer electrolyte was conducted at 60°C. How does this temperature affect the solubility and mobility of lithium salts within the electrolyte? Would performing the same experiments at room temperature (instead of 60°C) potentially yield different crystallization kinetics, and how might this influence the interpretation of the results?

Response: We want to thank the reviewer for this note. Please refer to our reply to comment #2 above.

Comment: While the manuscript presents valuable experimental techniques and observations regarding crystallization behavior in polymer electrolytes, it falls short in connecting these findings to the practical performance of single-ion conducting polymers. Furthermore, the testing conditions and lack of actionable insights significantly limit the study's relevance and impact. The authors need to:

Include electrochemical performance metrics for single-ion conducting polymers.
Clarify and validate how experimental conditions (temperature and pressure) affect the observed results.

Provide specific, constructive proposals for future research directions or strategies to mitigate crystallite formation.

In its current form, the manuscript is not recommended for publication in Nature Communications.

Response: Again, we want to thank the reviewer for acknowledging the results of our study and the high-value suggestions to help us improve the quality of the manuscript. We meet the reviewer demands by including all remarks in our revised manuscript. As described above, we implemented electrochemical performance tests to connect the influence of the crystallites on the ionic conductivity and transference number (see answer to comment #1), and we included a discussion about the influence of the experimental conditions (see answer to comment #2). Furthermore, we propose our understanding of the crystallization mechanism in the polymer electrolyte, and based on this, we derive strategies to mitigate the crystallization formation and propose future research directions (see answer to the comments #2 and #3).

Reply to Reviewer #2 comments to manuscript NCOMMS-24-74934:

Comment: I co-reviewed this manuscript with one of the reviewers who provided the listed reports. This is part of the Nature Communications initiative to facilitate training in peer review and to provide appropriate recognition for Early Career Researchers who co-review manuscripts.

Response: We thank the reviewer for co-reviewing our manuscript. The review helped us significantly to improve our work.

Reply to Reviewer #3 comments to manuscript NCOMMS-24-74934:

Comment: The manuscript delivered the observation of the metallic lithium and LiCO₃ crystal formation in the polymer electrolyte using the operando nano-focus WAXS at the early-stage cycling of the battery. A Li-symmetric CR2032-type coin cell equipped with PVDF-HFP and PSTFSILi single ion conductor was prepared and tested. The author highlights that the formation of metallic lithium inside the electrolyte might be correlated to the dendrite formation. According to the author, this technique has been rarely reported to be implemented on polymer lithium metal batteries. However, from the reviewer's point of view, the manuscript has not yet fulfilled the criteria to be published in a top journal, and is recommended to be submitted to a more specialized journal. The major issues are:

Response: We thank the reviewer for the critical feedback, which significantly helps us to improve the manuscript. We respectfully disagree with the statement that our work should be published in a more specialized journal. Dendritic lithium formation or generally crystallization inside the solid electrolyte, especially polymer electrolytes, is, to the best of our knowledge, a rather overlooked but extremely important degradation mechanism. Metallic lithium formation in the bulk of ceramic solid-state electrolytes has been observed by Han et al., published in *Nature Energy* (doi.org/10.1038/s41560-018-0312-z) and very recently by Liu et al., published in *Nature Materials* (doi.org/10.1038/s41563-024-02094-6), which shows its high relevance. Furthermore, the latter publication was additionally mentioned in a *Nature Materials News & Views* article (doi.org/10.1038/s41563-024-02104-7), which further demonstrates the importance of this topic. We extend the knowledge of dendritic lithium in the bulk from solid ceramic electrolytes to polymer electrolytes. Furthermore, we show the kinetics of Li₂CO₃ and LiOH crystallites inside the polymer electrolyte, which has not been shown yet, as this can be attributed to the dissolution of SEI compounds. With this in mind, we are convinced that with our work, an awareness of this degradation phenomenon can be raised. Therefore, mitigating the dissolution and formation of crystallites should be considered to build more long-lasting batteries. We are confident that with the extremely helpful comments and suggestions of the reviewer, which further strengthen the manuscript, the publication in this top journal is feasible.

Comment: 1. Lack of details and discussion about how the state-of-the-art nano-focus WAXS technique is implemented in the batteries cycling process, or in other words, what are the difficulties and specific innovations of this approach.

Response: We thank the reviewer for this remark, which helps us to clarify the uniqueness of our experimental approach. The beam size of X-rays and neutrons is typically quite large: for X-rays, ~ 50 - 400 μm, and neutrons, ~ several mm - cm. This makes a local investigation of one specific battery layer (for example, the polymer electrolyte) with scattering experiments very difficult and practically impossible. Therefore, in almost every battery *operando* transmission scattering experiment, the beam direction is parallel to the normal of the battery stack and hence every layer, meaning both electrodes and the electrolyte, are penetrated by X-rays or neutrons. Consequently, the resulting scattering signal contains information about all penetrated layers, and potential structure changes cannot be clearly assigned to one specific layer (for example, the polymer electrolyte). The specific innovation of using a nano-sized X-ray beam is that with the extremely small beam size of ~ 330 nm x 330 nm, the beam direction can be oriented parallel to the electrodes, and the aforementioned issue can be avoided. The X-ray beam only penetrates the polymer layer, and possible structure changes

can be clearly assigned to the polymer layer. For clarification, we added these sentences to the Supplementary Information.

It reads on page 4:

“The beam size of X-rays and neutrons is typically quite large: for X-rays, ~ 50 - 400 μm, and neutrons, ~ several mm - cm. This makes a local investigation of one specific battery layer (for example, the polymer electrolyte) with scattering experiments very difficult and practically impossible. Therefore, in almost every battery operando transmission scattering experiment, the beam direction is parallel to the normal of the battery stack and hence every layer, meaning both electrodes and the electrolyte, are penetrated by X-rays or neutrons. Consequently, the resulting scattering signal contains information about all penetrated layers, and potential structure changes cannot be clearly assigned to one specific layer (for example, the polymer electrolyte). The specific innovation of using a nano-sized X-ray beam is that with the extremely small beam size of ~ 330 nm x 330 nm, the beam direction can be oriented parallel to the electrodes, and the aforementioned issue can be avoided. The X-ray beam only penetrates the polymer layer; and possible structure changes can be clearly assigned to the polymer layer.”

Furthermore, we added a detailed illustrative comparison of both approaches to the Supplementary Information Fig. 2. It is shown in the Supplementary Information on page 3:

Comment: 2. Limited diversity in characterization methods and experimental validation to support the reliability of the observations. The study relies heavily on a single or narrow set of techniques, with limited experimental repetitions.

Response: We thank the reviewer for this very good point, which helps us to show the reliability of the nWAXS experiment. To validate our observations, we further performed X-ray independent FTIR and electrochemical measurements. FTIR measurements were conducted on the pristine polymer electrolyte and a polymer electrolyte film, which was placed directly after the electrolyte soaking step onto a lithium chip and was then in contact for eight days. The corresponding FTIR spectra of both polymers show clear differences, as additional peaks appeared in the FTIR spectrum of the polymer that was in contact with lithium. We can identify LiO-H vibrations, which can be attributed to LiOH. Furthermore, we see a clear increase in C=O bonds in the polymer that was contacted with lithium. This is a strong indication of carbonate groups in the polymer, which further underlines our statement that there is material dissolving from the lithium surface and transitions from the lithium side into the electrolyte. Furthermore, we took a photograph of the polymer electrolyte after eight days of contact, and a clear change in the surface of the polymer is visible by eye. The area that was in contact with the lithium chip appears to be black. This observation is in accordance with the paper by Shadike et al., who also showed a photograph of the SEI material that appeared black.

It reads on pages 5 and 6:

“To confirm the identity of these crystallites as lithium hydroxide and lithium carbonate, we further perform Fourier-transform infrared (FTIR) spectroscopy measurements on the pristine polymer and a polymer that was in contact with a lithium metal chip for eight days. The two FTIR spectra are shown in Fig. 1c, and a clear difference between these two spectra can be recognized as the contacted polymer shows a noticeable increase of stretching vibrations of the -C=O bond at $\nu(\text{C=O}) = 1775 \text{ cm}^{-1}$, which can be attributed to a rise in carbonate groups in the polymer.¹⁷ Furthermore, an emerging peak at 3566 cm^{-1} can be assigned to stretching vibrations of the LiO-H bond of $\text{LiOH} \cdot \text{H}_2\text{O}$ and is an indication of the presence of LiOH.¹⁷ The absence of any O-H bond vibrations in the pristine polymer film indicates that there is no H_2O contamination initially. Overall, the FTIR test shows two things: 1) The transition of compounds from lithium into the electrolyte is completely pressure independent and directly happens after contacting the materials at room temperature and 2) the peaks in the radial integrations in Fig. 1b are not X-ray induced and thus not a measurement artefact but indicate indeed crystalline compounds in the polymer reliably. Further identifiable peaks in the FTIR spectra are indicated and assigned in Fig. 1c correspondingly ($\nu(\text{C-H}) = 3020 \text{ cm}^{-1}$ and 2980 cm^{-1} , $\delta(\text{C-H}) = 1450 \text{ cm}^{-1}$ and 1480 cm^{-1} , and $\nu(\text{S-O}) = 1390 \text{ cm}^{-1}$).^{18,19} Furthermore, a photograph of the lithium-contacted polymer is given in Fig. 1d. The contacted area exhibits a discoloration of the polymer from white to black, which is clearly visible by eye and indicates the transition of compounds from the lithium metal surface into the polymer. This finding is in good agreement with the results by Shadike et al., who also showed a photograph of extracted SEI material that appeared black.²⁰”

It shows Fig. 1 on page 7:

Besides the FTIR measurements, we perform further electrochemical impedance spectroscopy measurements on two coin cells: the polymer electrolyte sandwiched between 1) symmetric stainless steel and 2) symmetric lithium cell. The idea behind this comparison is to see if the bulk ionic conductivity in the lithium symmetric cell is lower due to the presence of the

crystallites, as we proposed in the first draft. For that, we measure an impedance spectrum every 12 hours for 4 days and finally compare the resistances obtained from the Nyquist plots of both cells. As expected, both cells show the same trend in bulk resistance. However, we find that the bulk resistance obtained from the lithium symmetric cell is higher at every time compared to the symmetric stainless-steel cell (while having the same thickness). With this experiment, we can prove that the diffusion of lithium to form crystallites in the polymer electrolyte has a negative impact on the ionic conductivity of the polymer electrolyte. Furthermore, we conduct a chronoamperometry test combined with electrochemical impedance spectroscopy to estimate the lithium-ion transference number, which is around 0.68. Moreover, we conduct an electronic conductivity test according to Han et al. by applying 100 mV to a stainless steel symmetric cell, which yields a current of 0.8 nA. We have added this information.

It reads on pages 13 and 14:

“Next, we study the impact of the observed crystallites in the polymer electrolyte on its electrochemical properties. Therefore, electrochemical impedance spectroscopy measurements of this polymer electrolyte are performed with a stainless steel and lithium-symmetric cell over a time period of 96 h with 12 h steps. The corresponding Nyquist plots are depicted in Fig. 5a,b. The bulk resistance of the polymer is determined by the width of the (first) semicircle²³ and the extracted resistance values of both cells are compared in Fig. 5c. As expected, the trend of both resistance curves until equilibrium is the same, however the bulk resistance from the lithium-symmetric cell is around 20 Ω higher at every time compared to the stainless-steel symmetric cell. Furthermore, a transference number test is conducted by performing a combination of chronoamperometry with electrochemical impedance spectroscopy before and after polarization.²⁴ The width of the second semicircle, which represents the interfacial resistance, is used to estimate the transference number to be ~ 0.68 . Additionally, the electronic conductivity is determined by applying a voltage of 100 mV to a stainless-steel cell (blocking conditions) similar to Han et al.²⁵ A steady-state current of around $I = 0.8$ nA is set in after one hour. This current is low but non-zero and thus non-negligible.”

We added a new Figure (Fig. 5) with the corresponding results.

It shows Fig. 5 on page 15:

Comment: 3. The discussion primarily focuses on presenting results without diving into the probable underlying mechanisms of the local crystallization inside the electrolyte but not the interface.

Response: We thank the reviewer for this valuable remark. It was our clear intention to investigate possible crystallization in the polymer electrolyte bulk instead of crystallization at the interface. Regarding the mechanism, from our point of view, the compounds Li_2CO_3 and LiOH of the surface of the lithium metal are dissolved due to the contact with the electrolyte, especially due to the carbonates EC and PC, forming CO_3^- , OH^- , and Li^+ . We placed a polymer electrolyte film in contact with a lithium metal chip for eight days, and we can clearly see by eye a discoloration of the polymer from white to black. We added a photograph of this polymer film. Additionally, we perform FTIR measurement on this polymer film and the pristine polymer electrolyte, and can confirm the presence of LiOH and Li_2CO_3 . These ions can move in the solvent phase and to some extent in the amorphous phase of the polymer. Now, one possibility is to nucleate and form crystallites in the solvent phase. Another possibility is to accumulate at the crystalline phase, as this is a conducting barrier for the ions. As mentioned above, a very similar mechanism has just recently been proposed by Liu et al., published in *Nature Materials* (doi.org/10.1038/s41563-024-02094-6), where metallic lithium is formed by electrolyte reduction at grain boundaries. Based on this, we further performed electronic conductivity measurements in blocking conditions in a symmetric stainless steel cell by applying a DC voltage of 100 mV as proposed by Han et al. We find that the electronic conductivity of the polymer electrolyte is small but non-zero and thus non-negligible. Therefore, we conclude that, comparable to ceramic electrolytes, a reduction of Li^+ to metallic lithium is possible.

We added our understanding of the underlying mechanism accordingly.

It reads on pages 16 and 17:

“In general, the observed crystallites inside the polymer electrolyte can be responsible for the higher ionic bulk resistance of the lithium symmetric cell compared to the stainless-steel cell, as they can act as an ion conducting barrier (Fig. 5a,b,c). Therefore, the movement of the lithium ions can be hindered, especially as the ionic conductivity of bulk LiOH at room temperature is quite low²⁶. Li_2CO_3 and LiOH are an essential part of the SEI layer and the native passivation layer of the lithium metal electrode surface. Consequently, considering the solubility of Li_2CO_3 and LiOH in EC and PC²⁷, which is rather high compared to other SEI layer compounds, a possible explanation for the presence of these compounds can be given: Due to the contact of the lithium electrode with the soaked electrolyte film, the crystallites of the lithium surface can be dissolved, forming CO_3^- , OH^- , and Li , which diffuse into the electrolyte. This is validated by our FTIR measurements of the pristine and lithium-contacted polymer, which confirm the presence of Li_2CO_3 and LiOH compounds in a lithium-contacted polymer film. Then, the ions mainly move in the solvent phase, and one possibility is to nucleate and form crystallites in the solvent phase. Another possibility is to accumulate and crystallize at the interfaces between the amorphous and crystalline regions of the polymer electrolyte, similar to grain boundaries in ceramic electrolytes or at defects. The nWAXS results suggest that the lithium carbonate crystallites either move and thus change their position or be (partially) dissolved and re-crystallize at a different position, in contrast to the LiOH crystallites, which tend to stay stable over time. Additionally, the presence of the anions CO_3^- and OH^- can be responsible for lowering the total transference number. Therefore, it is of high importance to mitigate the dissolution of these compounds. This can be achieved, for example, by coating the lithium surface with a non-soluble layer with a sufficiently high ionic conductivity. Besides surface engineering, the general development of electrolytes and their additives should take the solubility of Li_2CO_3 , LiOH , and other SEI compounds into account. Therefore, we propose to include corresponding experimental tests in their characterization. On top of that, we observe the early stages of the formation of pure metallic lithium inside the polymer electrolyte, which is not necessarily connected to the lithium metal surface but can lead to dendritic structures. Metallic lithium formation in the bulk of ceramic solid-state electrolytes has been observed by Han et al.²⁵ and very recently by Liu et al.²⁸ due to the

reduction of lithium ions. Our study expands such observations to polycrystalline gel-type polymers, which typically contain PVDF-HFP and EC/PC. The formation of metallic lithium in the polymer bulk can be caused by a non-negligible electronic conductivity (Fig. 5e) and the resulting reduction of Li^+ . This result is an important expansion of the current understanding of dendritic lithium growth in polymer electrolytes, which is thought to originate from the lithium metal surface.²⁹

Comment: Some other comments and questions are listed below.

1. Figure or Fig.? This should be uniform in the manuscript, and the figure number should be given clearly. For example, Line 168, which figure is being discussed? Line 189, which figure in supplementary?

Response: We thank the reviewer for this very attentive observation. We changed all “Figures” to “Fig.” in the manuscript.

Former lines 168 and 189: We added the radial integrations of the high q-values to see the LiOH peaks. It is now Fig. 3 in the revised version (see remark #4).

Comment: 2. Some typo errors: Line 148 Fig. 2c-f, Fig. 2c-e? Line 172, the word H20, H2O?

Response: Again, we thank the reviewer for spotting these mistakes, which improve the accuracy of our manuscript. As we added a third radial integration (reviewer comment #4), we now have six subplots in Figs. 3, and 4,5, meaning from a-f, which was Fig. 2 before. We changed the text accordingly.

Moreover, we corrected the typo in the word H₂O.

It reads on page 5:

“The origin of LiOH might stem from H₂O contaminations; however, as the peaks are not present in the static measurement, a formation of LiOH crystallites in the electrolyte due to the contact of the polymer layer with the lithium metal electrodes is more likely.”

Comment: 3. Regarding the discussion in Line 188-192 and Supplementary Figure 4, it seems the peak of $q \sim 2.47 \text{ \AA}^{-1}$ is closer to that of LiF. It seems it is possible to have two species simultaneously. Since there is a time difference during the scan as discussed by the author, is there any influence on the result? Any comment on this?

Response: We thank the reviewer for this very helpful remark, which helps us clarify the interpretation of the results. We re-analyzed the data and are still convinced that we have LiOH in the polymer. Especially, the positions and also the distances of the two measured peaks at $q \sim 2.25 \text{ \AA}^{-1}$ and $q \sim 2.47 \text{ \AA}^{-1}$ match very well with the theoretical ones. Additionally, the reflexes at higher q-values match the theoretical ones of LiOH very well. For clarity, we deleted Figure 4 in the Supplementary Information and added the radial integrations of this q-range and the corresponding theoretical reflexes of LiOH and LiF to Fig. 3,a,b,c. This representation is more comprehensive and indicates that the peaks are clearly identified as LiOH.

It reads on page 10:

“Furthermore, higher orders of LiOH at $q \sim 3.34 \text{ \AA}^{-1}$, $q \sim 3.47 \text{ \AA}^{-1}$, and $q \sim 3.78 \text{ \AA}^{-1}$ are identified. Lithium fluoride (LiF, mp-1009009) also has peaks theoretically located at $q \sim 2.47 \text{ \AA}^{-1}$ and $q \sim 3.47 \text{ \AA}^{-1}$. However, the ratio of the experimental intensities for example at $q \sim 3.47 \text{ \AA}^{-1}$ and $q \sim 3.78 \text{ \AA}^{-1}$ matches very well with the theoretical ratio of the LiOH peaks (~ 1.17 vs.

~1.13), which suggests that the most intense theoretical LiF peak at $q \sim 3.47 \text{ \AA}^{-1}$ is not present and thus, also the peak at $q \sim 2.47 \text{ \AA}^{-1}$ cannot be assigned to LiF. As described by Tan et al., the formation of LiF needs the presence of an NMC811 cathode material and a high voltage to produce LiF at least in the SEI.¹⁵“

The new Figure of the corresponding radial integrations is shown in the response to the next comment.

Comment: 4. In Figure 2 and Figure 4, why only two half cycles of radial integrations are given?

Response: We thank the reviewer for this remark. We decided to give only two radial integrations to show one without and one with an additional peak. However, since this might raise confusion for the reader, we added a third radial integration meaning for every half cycle one representational radial integration.

It shows Figs. 2-4 on pages 9, 11, and 13:

Comment: 5. How many cells have been tested? Is it a common phenomenon of the formation of metallic lithium and Li_2CO_3 crystals inside the polymer electrolyte in such a battery system?

Response: We thank the reviewer for this comment. We exclusively tested this polymer electrolyte as its materials (especially PVDF-HFP, EC, PC) are widely-used in (commercial) battery applications. Besides X-ray measurements, we complement the analysis of the polymer electrolyte by electrochemical and FTIR measurements (see answer to comment #2). Therefore, for any polycrystalline gel-type polymer electrolyte swollen in EC/PC, that has a non-zero electronic conductivity, we expect the formation of metallic lithium, Li_2CO_3 , and LiOH crystallites inside the polymer. Overall, this means that three different experimental approaches (X-ray, FTIR and electrochemistry) indicate the presence of the compounds in the polymer electrolyte (see above). We added this point into the discussion part.

It reads on page 17:

“Our study expands such observations to polycrystalline gel-type polymers, which typically contain PVDF-HFP and EC/PC.”

Comment: 6. If this work focuses on tackling the technical issues of applying the operando nano-focus WAXS on the polymer electrolyte battery cell, more details and discussion about the approach should be given. If the point is to give new insight for exploring the SEI and dendrite formation in polymer electrolyte batteries, more characterizations and experimental validations that support the observation should be done. Additionally, it is necessary to provide more discussions on the mechanism and the dynamics process of local metallic lithium formation. For example, what are the mainstream viewpoints on the lithium dendrite formation process in polymer batteries, and what is the impact of the observation in this work on these viewpoints? And when the dendrite forms, whether the failure of the battery could be observed.

Response: We thank the reviewer for this remark. We further performed FTIR measurements and tested the polymer electrolyte electrochemically (see reply to comment #2).

The mainstream viewpoint of lithium dendrite growth is that this growth originates from the lithium surface. We added this information to the main text.

It reads on page 17:

“This result is an important expansion of the current understanding of dendritic lithium growth in polymer electrolytes, which is thought to originate from the lithium metal surface.²⁹”

Last point: We are convinced that the dendritic structures can be observed with nWAXS after the battery is shorted. For that, we propose post-mortem *ex-situ* scanning nWAXS experiments: The battery can be cycled in typical cell formats such as coin cells or pouch cells. After the cell has shortened, the electrolyte will be extracted. Then, in a scanning nWAXS experiment, the electrolyte can be investigated by the nano-sized X-ray beam in the same manner as in this experiment. We have added this suggestion to the revised manuscript.

It reads on page 17:

“To test the influence of the lithium crystallites on the long-term stability, we propose post-mortem ex-situ scanning nWAXS experiments as a future research direction. Several lithium (symmetric) cells with various electrolytes, surface coatings, or temperatures, and hence different lifetimes, can be investigated after the cells have been shortened. The extracted electrolytes can be investigated by scanning nWAXS in the same manner as in this study. Thus, possible correlations between lifetime and internal lithium crystallization can be explored. Especially higher temperatures can have an impact on the lithium crystallization process as reaction kinetics and solvent decomposition can be altered.”

Reply to Reviewer #4 comments to manuscript NCOMMS-24-74934:

Comment: Pr Muller-Buschbaum and colleagues report nanofocus WAXS to trace spatially resolved occurrence and development of crystalline species at the interface between a cycling Li metal electrode and a polymer electrolyte. They identify Li_2CO_3 , LiOH and Li metal at various distances from the bulk metal surface.

The interesting point of this study is that the presence and temporal evolution of crystalline compounds such as the mentioned ones can (potentially) be traced as a function of distance from an electrode during cycling. However, the key weaknesses which really need to be addressed before further consideration are

A) The experimental setup is too poorly described as to really be able to understand what is measured.

B) The scholarly presentation is very poor, seriously hampers understanding, and reminds me of an early draft. I even wonder if all authors read the manuscript as this was not caught.

C) What is the scientific insight (beyond showing the methodological possibility of measuring those crystalline compounds in time and space)?

Response: We thank the reviewer for acknowledging the results of our study and the high-value suggestions to help us improve the quality of the manuscript. Regarding points A) and B), we are confident that with the feedback of the reviewer, we are able to enhance the quality of the scholarly presentation. Regarding point C), the scientific insight is mainly that we observe crystallites in the polymer electrolyte, which originate from the lithium metal surface. These crystallites form immediately by contacting the polymer and lithium. Consequently, these crystallites lower the ionic conductivity of the polymer and hence can worsen its electrochemical performance. We added it to the discussion part accordingly.

It reads on page 16:

“In general, the observed crystallites inside the polymer electrolyte can be responsible for the higher ionic bulk resistance of the lithium symmetric cell compared to the stainless-steel cell, as they can act as an ion conducting barrier (Fig. 5a,b,c). Therefore, the movement of the lithium ions can be hindered, especially as the ionic conductivity of bulk LiOH at room temperature is quite low²⁶. Li_2CO_3 and LiOH are an essential part of the SEI layer and the native passivation layer of the lithium metal electrode surface. Consequently, considering the solubility of Li_2CO_3 and LiOH in EC and PC²⁷, which is rather high compared to other SEI layer compounds, a possible explanation for the presence of these compounds can be given: Due to the contact of the lithium electrode with the soaked electrolyte film, the crystallites of the lithium surface can be dissolved, forming CO_3^- , OH^- , and Li, which diffuse into the electrolyte. This is validated by our FTIR measurements of the pristine and lithium-contacted polymer, which confirms the presence of Li_2CO_3 and LiOH compounds in a lithium-contacted polymer film.”

Comment: Here are specific points further substantiating A) and B)

1) The research question is very diffuse and not clearly discernable from the abstract and intro, and hence, the learning is equally diffuse (C above). I warmly recommend this book, which shows how this could be tackled <https://global.oup.com/academic/product/writing-science-9780199760244?cc=at&lang=en&>. The abstract mentions as motivation for the work “... the practical usage can be debated”, which leaves the reader wondering why nWAXS could help. What does “Surprisingly, we observe the local kinetics of lithium carbonate,

lithium hydroxide, and metallic lithium crystallites in the polymer electrolyte in the vicinity of the lithium electrode during three half-cycles over a depth of approximately 16 μm measured from the electrode” mean? Wasn’t this the purpose? What is kinetics of crystallites? The introduction is with ~1000 words way too long (double of what it should be), wanders around in a review, and doesn’t come to the point either. In my view it could be as simple as saying: We want to measure crystalline compounds in the electrolytes in-situ, with ~micron resolution up to some 10 micron away from the electrode and show by means of a polymer electrolyte on Li metal that this is useful. I’d buy that.

Response: We thank the reviewer for the constructive feedback and the valuable suggestions to help us improve the comprehensibility of our manuscripts. We highly appreciate the book recommendation and purchased it accordingly (see the following photograph).

We agree with the reviewer that the introduction was too long and diffuse. Therefore, we cut it in half and rephrased it to be more concise.

It reads on pages 3 and 4:

“Lithium metal is regarded as the ideal anode for lithium batteries as it exhibits the lowest electrochemical potential (-3.04 V vs. standard hydrogen electrode) and the highest specific capacity (3860 mAh g⁻¹).¹ However, the combination of lithium metal with conventional, non-aqueous liquid electrolyte (e.g., carbonate-based) is accompanied by uneven dendritic Li growth on the surface of the anode, which leads to rapid performance loss, and even the explosion of the cells is possible. Besides tuning the composition of the electrolyte through additives² or passivating the electrode’s surfaces³, the use of (solid) polymer electrolytes is a strategy to bypass the aforementioned issues due to superior thermal, electrochemical, and mechanical stability.⁴⁻⁶ Among the variety of polymer electrolytes, single-ion conducting polymers are an interesting group as they exhibit a transference number close to unity due to the tethering of the anion to the polymeric backbone. According to theoretical models⁷, this property is accompanied by the suppression of dendritic structures on the lithium metal surface. However, the usage of single-ion conducting polymers is debatable as it is reported that the benefits come only into play at impractically high currents and temperatures.⁸ Therefore, the major goal of this study is to test if a single-ion conducting polymer can indeed suppress lithium dendrite formation and, if not, to observe where the dendrites form, i.e., if the picture that dendrite growth starts at the electrode is correct. For that reason, we make use of synchrotron-based nano-focus wide-angle X-ray scattering (nWAXS), as the nanometer-sized X-ray beam allows for scanning locally only the polymer electrolyte in the electrode-near area, in an orientation of the cell that prevents getting background from the electrodes. Using nWAXS, a high spatial resolution of the crystalline structure can be provided⁹, and possible structures that are hardly visible with real space microscopy techniques can be resolved. Hence, the formation of possible lithium crystallites can be detected at the electrodes or inside the polymer-based electrolyte. Furthermore, in combination with the operando set-up,

the temporal resolution is also achieved, and thus, physical-chemical processes can be studied.

In this work, we perform operando scanning nWAXS at room temperature on a symmetric lithium cell with a polymer-based electrolyte (PVDF-HFP/PSTFSiLi in a mixture of ethylene carbonate and propylene carbonate (EC/PC)), which is specially designed for synchrotron experiments to spatially and temporally resolve the crystalline structure of the electrolyte on a nanoscale. With such an approach, we are able to identify rare crystallization events in the early stages of cell operation. Besides detecting metallic lithium in the polymer electrolyte, we surprisingly observe the unexpected local kinetics of lithium carbonate and lithium hydroxide in the vicinity of the lithium electrode during three half-cycles over a depth of approximately 16 μm measured from the electrode.”

Comment: 2) The “special cell”, which is of course pivotal to perform this study is not at all explained at any reasonable depth of detail. Fig. S1 is, if I understand right, the transmission cell, where all layers are penetrated. Different from the “special capillary cell” mentioned in l. 70 of the intro. Or was the cell stack parallel to the screw faces and hence beam, which come from top and bottom into the cell stack? If so, the in-plane dimension was of the cell was very large (4 mm or so like the M4 screws?). Or is it 5 mm as suggested in l. 139? What is the purpose of the glass capillary seen in Fig. S2b? Why would it not be possible to heat this cell to 60 °C? That doesn’t appear too difficult. But the central question around this setup is (if I understood the setup right as mentioned) that your beam goes parallel to Li metal through ~4-5 mm of the polymer with 0 to 16 micron distance. How do you get the Li metal to a planeness of ~ micron and this plane with this precision parallel to the beam? If the Li metal surface has height variations of only

Response: We thank the reviewer for these remarks and want to clarify the experimental cell used. Yes, the used transmission cell was depicted in the original draft in Fig. S1 and is different from the special capillary cell, which was used in a work by Moehl et al in 2018, who first performed X-ray transmission scattering experiments on a polymer electrolyte (not used in this experiment). We deleted the part about the special capillary cell from Moehl et al. to reduce confusion. We added a schematic of the used cell and the corresponding X-ray path in Supplementary Fig. 2.

It reads in the Supplementary Information on page 4:

“The beam size of X-rays and neutrons is typically quite large: for X-rays, ~ 50 - 400 μm , and neutrons, ~ several mm - cm. This makes a local investigation of one specific battery layer (for example, the polymer electrolyte) with scattering experiments very difficult and practically impossible. Therefore, in almost every battery operando transmission scattering experiment, the beam direction is parallel to the normal of the battery stack and hence every layer, meaning both electrodes and the electrolyte, are penetrated by X-rays or neutrons. Consequently, the resulting scattering signal contains information about all penetrated layers, and potential structure changes cannot be clearly assigned to one specific layer (for example, the polymer electrolyte). The specific innovation of using a nano-sized X-ray beam is that with the extremely small beam size of ~ 330 nm x 330 nm, the beam direction can be oriented parallel to the electrodes, and the aforementioned issue can be avoided. The X-ray beam only penetrates the polymer layer, and possible structure changes can be clearly assigned to the polymer layer.”

It shows in the Supplementary information on page 3:

The glass capillary in Fig. S2b is the direct beamstop, which protects the detector from the direct beam and is independent from the electrochemical cell. For clarity, we added the word “direct beamstop” into the figure.

It shows in the Supplementary Information on page 5:

Our intention was to perform this experiment at room temperature to have real-world conditions.

Yes, the X-ray beam is oriented parallel to the lithium metal electrode and propagates only through the polymer electrolyte. By adding flat copper platelets between the screw and the lithium chip, we flatten the lithium electrode. Furthermore, we align the X-ray beam in a way to have a clear scattering signal of the polymer.

It reads on page 19:

“Additionally, flat copper chips were placed between the lithium metal and the screws to protect the soft lithium metal and prevent short circuits. Furthermore, they guarantee sufficient planeness of the lithium chip.”

Comment: 3) Fig. 2: it is not at all clear what $x = 0 \dots 4$ mean. Also you mention that “curves are shifted along the y axis for clarity of the presentation”, y-axis of the scanning area or the abscissa of the plot? At what y height above the Li metal if this is meant by the sketch in c? Fig 2cde: what does it mean “combined with EC data”? the map is measured after 0.5, 1, 1.5 h in c,d,e? or during this 1/2h each? L. 148: there is no Fig. 2f. At what point has Fig. 1b been measured?

Response: We thank the reviewer for these remarks and we apologize for the confusion caused by the very short description. $X = 0 \dots 4$ means the width of the scanned area. We added $4 \mu\text{m}$ with arrows instead of just writing $x = 0 \dots 4$ to enhance the clarity. See for example or reply to comment #5.

The five curves (radial integrations) are shifted along the abscissa of the plot as otherwise they would overlap. Hence, with this it is not meant the height y above the lithium metal by the sketch in c. We clarify this in the revised manuscript.

It reads on page 8:

“For clarity, the five radial integrations of each half-cycle are shifted along the abscissa of the plot as they would otherwise overlap.”

The map is measured during (*operando*) each half-hour plating/stripping process, meaning that while the electrochemical cell is running, the map is recorded. That is why we plotted the voltage curve vertically to make clear that the time is proceeding and the area is scanned by the beam. We clarified this by adding a schematic of the scanned area next to every 2D map.

The GIWAXS measurement of Fig. 1b (comparable to an XRD) was performed independently from the nano-focus experiment and only on the pristine polymer film without contact to lithium metal. We added a photograph of the used sample environment, which provides an argon atmosphere, to the Supplementary Information. It shows in the Supplementary Information on page 2:

Comment: 4) If Li_2CO_3 , mp 556777 means the materials project entry, it should be written as Li_2CO_3 , mp-556777.

Response: We thank the reviewer for this remark and have changed it accordingly in the manuscript.

It reads on page 5:

“This peak matches quite well with the first Bragg peak of lithium carbonate (Li_2CO_3 , mp-556777), however no further Li_2CO_3 peaks can be identified.”

Comment: 5) L. 186: where are these broad peaks at 22.25 and 2.47 Å⁻¹ seen? In Fig S4? But at what point has it been measured? No indication in the At what point has Fig. 1b been measured? No such peaks there.

Response: We thank the reviewer for this remark. These two broad peaks are permanently present in the area that we measured. We show in the revised version in Fig. 1b a rough, vertical scan of the polymer before the experiment, and see that these two broad peaks are present closer to the interfaces but not in the bulk of the polymer. The operando scanning experiment was then performed in the area near the lithium electrode where these two peaks are permanently present. We added selected radial integrations from all three half-cycles. It shows on page 11:

The GIWAXS measurement of Fig. 1b (comparable to an XRD) was performed independently from the nano-focus experiment and only on the pristine polymer film without contact to lithium metal. We added a photograph of the sample environment, which provides an argon atmosphere to the supplementary information (see reply to comment #3).

Exactly! The peaks are not present in the GIWAXS measurement of the pristine polymer film. However, in contact with the lithium metal, we find that these peaks are present in the nWAXS experiment. This fact lets us conclude that these originate from the diffusion of LiOH

from the lithium metal into the polymer electrolyte. We can further validate this by performing FTIR measurements on a pristine polymer and one, which was in contact with lithium for eight days. We added these results in a new Fig. 1:

It shows Fig. 1 on page 7:

Comment: 6) What does “kinetics of these crystallites are quite limited” in l. 183 mean? What is kinetics of crystallites? They move spacially or grow/shrink over time?

Response: We thank the reviewer for this remark. With the limited kinetics, we intend to express that there is hardly any change in the LiOH crystallite intensity over time. That means that these crystallites do not move, dissolve, or recrystallize (= kinetics) in an extensive manner. We clarify this in the revised manuscript.

It reads on page 10:

“This finding suggests that the kinetics of these crystallites are quite limited, meaning that there is hardly any change in the LiOH crystallite intensity over time. This finding can be understood as the fact that these crystallites do not move, dissolve, or recrystallize (= kinetics) in an extensive manner [...]”

Comment: 7) L. 211: “The observation of lithium crystallites matches quite well with the previously mentioned appearance of isolated lithium metal inside the polymer electrolyte”. Where was this mentioned?

Response: We thank the reviewer for this remark. In lines 63-65 of the original draft, we refer to a study in *Nature Energy* (doi.org/10.1038/s41560-018-0312-z), which could prove the formation of metallic lithium inside ceramic electrolytes due to sufficient electronic conductivity of the electrolyte. We shifted this from the introduction to the discussion part and cited it directly.

It reads on page 17:

“Metallic lithium formation in the bulk of ceramic solid-state electrolytes has been observed by Han et al.²⁵ and very recently by Liu et al.²⁸ due to the reduction of lithium ions.”

Comment: 8) Fig. S5: are these non-identified peaks only at certain spots, do they evolve over time? Are they present initially?

Response: We thank the reviewer for this remark. The non-identified peaks are not present initially. For clarity, we shifted the vertical scan of the polymer before the operando experiment to the main text to make clear that these peaks are not initially present.

It reads on page 17:

“The fact that also not identifiable reflexes show up during cell cycling, which are not initially present and only appear at certain spots rarely, underlines the complexity and uniqueness of the interplay between the electrolyte and the lithium metal and the corresponding SEI compounds.”

Comment: 9) Discussion: “However, our result of assigning the peaks to LiOH can be substantiated by Tan et al., who describe the necessity of an NMC811 cathode material and the high voltage to produce LiF at least in the SEI”. Which paper you mean? Shouldn't it be cited here? What does this sentence mean? NMC811 produces LiF at high V. But how is this connected with LiOH at Li metal? L. 236: “SEI layer elements” elements or compounds? L. 243: what do you mean with “complexity and uniqueness of the system”? uniqueness that the present cell maybe is not representative of a non-operando cell with the same components? Is it then useful?

Response: We thank the reviewer for these remarks.

As the theoretical peaks of LiOH and LiF share almost the same q -positions, it can be tricky to distinguish which compound we have inside the polymer (see reply to comment #5). We come to the conclusion that, by comparing the intensities of the measured peaks, we have LiOH instead of LiF. We want to back up our claim by referring to the paper of Tan et al. (10.1038/s41565-022-01273-3). In this work, it was found that the formation of LiF, at least in the SEI, needs a high-voltage cathode; otherwise, there is no LiF formation. We clarify this in the revised manuscript.

It reads on page 10:

“Furthermore, higher orders of LiOH at $q \sim 3.34 \text{ \AA}^{-1}$, $q \sim 3.47 \text{ \AA}^{-1}$, and $q \sim 3.78 \text{ \AA}^{-1}$ are identified. Lithium fluoride (LiF, mp-1009009) also has peaks theoretically located at $q \sim 2.47 \text{ \AA}^{-1}$ and $q \sim 3.47 \text{ \AA}^{-1}$. However, the ratio of the experimental intensities for example at $q \sim 3.47 \text{ \AA}^{-1}$ and $q \sim 3.78 \text{ \AA}^{-1}$ matches very well with the theoretical ratio of the LiOH peaks (~ 1.17 vs. ~ 1.13), which suggests that the most intense theoretical LiF peak at $q \sim 3.47 \text{ \AA}^{-1}$ is not present and thus, also the peak at $q \sim 2.47 \text{ \AA}^{-1}$ cannot be assigned to LiF. As described by Tan et al., the formation of LiF needs the presence of an NMC811 cathode material and a high voltage to produce LiF at least in the SEI.¹⁵”

We mean SEI compounds.

It reads on page 16:

“Considering the solubility of Li_2CO_3 and LiOH in EC and PC²³, which is rather high compared to other SEI layer compounds, a possible explanation for the presence of these compounds can be given.”

With the wording “complexity and uniqueness of the system”, we refer to the stack of the electrolyte, the lithium metal, and the resulting dissolution of the complex SEI layer compounds. We reworded the sentence to be more precise.

It reads on page 17:

“The fact that also not identifiable reflexes show up during cell cycling, which are not initially present and only appear at certain spots rarely, underlines the complexity and uniqueness of the interplay between the electrolyte and the lithium metal and the corresponding SEI compounds (Supplementary Fig. 5).”

Reply to Reviewer #1 comments to manuscript NCOMMS-24-74934A-Z:

The authors' thorough and constructive responses to the initial round of reviews. The revised manuscript has significantly improved in both scientific rigor and clarity. In particular, the authors have made commendable efforts to address the core concerns raised in the initial review. They have effectively correlated the observed crystallization phenomena with electrochemical performance metrics, clarified how experimental conditions such as temperature and pressure influence their findings, corrected and refined material terminology and phase assignments, and proposed specific strategies and future directions to mitigate crystallite formation in polymer electrolytes. The inclusion of EIS, transference number, and electronic conductivity measurements now provides a clear link between the observed crystallites and their impact on ionic conductivity. This strengthens the practical relevance of the operando nWAXS results. The discussion section is now better structured, with more insightful mechanistic interpretations and clearer language.

Specific Comments

1. Electrochemical Performance Integration

The added impedance, transference number, and electronic conductivity measurements are well executed and convincingly support the negative impact of crystallite formation on transport properties. The comparison between symmetric stainless steel and Li-Li cells is especially helpful.

2. Clarification on Experimental Conditions

The authors provided a reasonable rationale for conducting measurements at room temperature and have included a forward-looking plan to explore high-temperature behavior. Similarly, the clarification that no significant pressure was applied during cell assembly is well received.

3. Terminology Precision

The revised abstract and main text now correctly distinguish between PVDF-HFP and PSTFSILi, accurately attributing single-ion conductivity to the latter. This correction is important and appreciated.

4. Phase Identification of Li_2CO_3 and LiOH

The FTIR data and updated high-q radial integrations provide compelling support for the presence of LiOH rather than LiF. While complementary XAS/NMR would further enhance confidence, the authors acknowledge this limitation and make a sound case using available data.

5. Mechanistic Insight

The revised discussion offers a more coherent and insightful interpretation of the spatial redistribution and persistence of different crystalline phases. The analogy to grain boundary reduction in ceramics is particularly interesting and expands the relevance of the findings.

6. Improved Presentation

The manuscript has been revised for clarity and conciseness, particularly in the introduction. The explanations of the scanning geometry, beam alignment, and sample flatness now resolve previous ambiguities.

In conclusion, the authors have fully addressed my concerns and substantially improved the manuscript. I now support publication in *Nature Communications*.

Response: We thank the reviewer for this wonderful comment. We truly acknowledge the effort for helping us to improve our manuscript and we are happy that the reviewer agrees for publishing our work in *Nature Communications*.

Reply to Reviewer #2 comments to manuscript NCOMMS-24-74934A-Z:

Response: We thank the reviewer for helping us to improve our manuscript.

Reply to Reviewer #3 comments to manuscript NCOMMS-24-74934A-Z:

The authors have thoroughly addressed the concerns. The manuscript may be published in its current form.

Response: We thank the reviewer for helping us to improve our manuscript. We are happy that the reviewer agrees to publish our work in *Nature Communications*.

Reply to Reviewer #4 comments to manuscript NCOMMS-24-74934A-Z:

The authors did make efforts to address the reviewer comments. The initially poor scholarly presentation improved a lot, but should still be improved. Methods need clarifications.

Response: We thank the reviewer for acknowledging our efforts to improve the manuscript.

1) Experimental setup: I still don't get it how this far-from-precision-engineered and machined cell would allow a flatness of Li metal surfaces of <1 micron over the size of ~5 mm. I do have extensive experience with precision engineering and know well the limits of certain precisions (flatness, parallelity, parallel movement). You really want to say that two M5 screws that are screwed from the two sides into a plastic block have a parallelity to allow for a 350 nm beam passing in parallel to a Li metal plate less than 4 micron for 5 mm? These screws are sealed with a teflon tape. First, this is already for sealing purposes not good practice as sealing a tread with Teflon tape is done only for tapered threads where the gap between the male and female thread closes in as it is screwed in and this way it progressively seals. Doing the same on a parallel thread is "Pfusch". Second, this way the threads are definitely not kept precisely axial (which a thread is not meant for anyway). The front faces of the screws are not planar with precision. At least this is what the SI Fig. 2c shows. Also, flat Cu plates will not make the Li/polymer interface more flat than it initially was. So how did you get the Li having less than a few micron deviation from planarity over the few mm? also the Cu is not shown in Fig. S2b. How did you determine the zero position above Li? How did you align the in-plane direction of the cell parallel to the beam? Was the cell on a z, tip, tilt stage?

Fig S2b is not helpful beyond a rough sketch. Why don't you make a standard construction drawing with real sizes of your components in the region of the cell stack, i.e. the screws, the Cu, Li, polymer? What you want to hide?

Response: We thank the reviewer for being critical about our setup and acknowledge the expertise in precision engineering, which helps us to better explain the details of the experiment. We are aware of the precision limits of the used cell and the battery stack. We do not claim to have a perfectly flat lithium metal. We focus the study on the polymer itself and not particularly on the Li/polymer interface, as we measure the 16 μm "somewhere" inside the polymer (please also see below), sufficiently above the lithium. Therefore, the exact position above the lithium is secondary. A possible tilt was aligned with the precision of the nano-sized X-ray beam, which is essential to avoid any scattering contributions from other parts. As an analogy, one can imagine the ratio of the nano-sized beam (diameter ~ 350 nm) through the polymer (thickness ~ 80 μm) similar to a rice grain (diameter ~ 2 mm) "shooting through" two piled up 0.5-liter water bottles (height = 2 * ~ 23 cm = 46 cm). We clarified it accordingly in the following.

It reads on page 4:

"Besides detecting metallic lithium in the polymer electrolyte, we observe the unexpected local kinetics of lithium carbonate and lithium hydroxide inside the polymer during three half-cycles over a depth of approximately 16 μm ."

As the limits with flatness and parallelism are well-described by the reviewer, we did not align specifically for measurements at the Li/polymer interface but for measurements in the polymer bulk being independent from the perfectness of the Li metal surface. Measuring the interface Li/polymer would require a different approach in sample preparation, including for

example, deposition techniques such as sputtering and physical vapor deposition of copper and lithium on, e.g., doped silicon, which would result in a different type of battery, namely a thin-film battery. However, for the purpose of our experiment, namely to see what is happening inside the polymer electrolyte, speaking already more about bulk properties, the cell stack with real-world-sized materials is sufficient, as we can confirm with complementary FTIR measurements.

Furthermore, we are also aware of the unconventional screw sealing with Teflon tape, which is a consequence of the lithium metal's sensitivity towards moisture. Therefore, we had to do everything possible to seal the cell and protect the lithium as much as possible during the synchrotron beamtime. The reviewer might call it “Pfusch” and not good practice; however, the stable electrochemical performance of the cell at the synchrotron proves us right. The plateau behavior indicates a good “state of health” of the materials over time, and an insufficiently sealed cell would result in a messed-up time-voltage curve being far away from a plateau. We have collected extensive experience with electrochemically non-well-working cells and know how chaotic their electrochemical curves can look like.

It reads on page 19:

“The threads of the two screws from the top and bottom were wrapped with Teflon tape to protect the samples from moisture degradation,”

Furthermore, we chose a relatively small horizontally scanning direction (4 μm of a total 7.5 mm) to keep any effects induced by a possible horizontal tilt at a minimum. The alignment was done in a horizontal and vertical way to have a clear transmission signal of the polymer (broad halo).

It reads on page 20:

“A rough vertical scan along the cell stack was performed to identify each layer. The nano beam was aligned vertically and horizontally for the polymer bulk (but not the Li/polymer interface) to obtain a clear transmission signal with sufficient distance from the electrodes.”

And:

“Furthermore, a relatively small horizontally scanning direction (4 μm of total 7.5 mm) was chosen to keep any effects induced by a possible residual horizontal tilt at a minimum.”

We can definitely deny the reviewers' assumption about us hiding any features as this is not the case. We intend to act as transparent as possible in the sense of good scientific practice. In addition to the rough sketch, we added a standard construction drawing accordingly to the Supplementary Information.

It shows on page 6 of the Supplementary Information:

And it reads on page 19:

“For more details, a construction drawing is given in Supplementary Fig. 3.”

Furthermore, it shows on page 5 of the Supplementary Information:

Overall, we observe crystallites and their behavior in the polymer electrolyte with the nWAXS experiment. As we can confirm this finding with complementary, completely independent, FTIR measurements of the polymer itself, we can be sure that we measured correctly despite the non-precision engineered cell. We thereby do not necessarily need to have a precision-engineered setup, as it is sufficient to reveal general crystallite behavior inside the electrolyte, but not specifically at the interface. We are now confident that we have made our point clearer with the reviewers' suggestions.

2) Does Fig 2 and the others where positions relative to Li metal are shown really mean, that the lowest point is < 1micron above the Li? Or do you simply mean that these 16 micron height scan are somewhere inside the polymer, sufficiently above the Li? But the methods at 1.342 claim that "he mesh grid size covered 4 x 16 μm^2 (HxV) with 5 x 17 points, pronouncing the vertical direction of the polymer electrolyte with respect to the electrode". As discussed above, it appears impossible that you can have such a precision above the Li.

Response: We thank the reviewer for this observation. Yes, as mentioned above, we simply mean that these 16 μm are "somewhere" inside the polymer, sufficiently above the lithium. If we would have intended to focus on the interface (but as we discussed above, the cells' design

is limited for that due to flatness and parallelism reasons), we would have chosen a different approach in sample preparation and also in the measurement protocol, emphasizing the horizontal direction far more. That is what we want to express with “pronouncing the vertical direction”. We deleted “with respect to the substrate”

It reads on page 20:

“The mesh grid size covered $4 \times 16 \mu\text{m}^2$ (HxV) with 5×17 points, pronouncing the vertical direction of the polymer electrolyte”

Overall, as the title says, the focus of the study is the polymer itself and not particularly the Li/polymer interface. Otherwise, we would have included the Li/polymer interface in the title. We use the small beam size to make sure to locally measure the polymer and not average over large areas as commonly done with big beam sizes. The schematic is just a rough illustration for understanding, but we can see that this might induce confusion so that we adjusted it. Specifically, we shifted the “scanned polymer area” upwards in Fig. 2,3, and 4, and adjusted the surface of the lithium electrode.

It shows now, for example, on page 9:

3) The use of the word “kinetics” is still questionable. Kinetics has a meaning in physics, i.e. the proportionality of a flux in response to a driving force. But here it is simply used for a temporal change. This is the wrong use of the word and is confusing.

Response: We thank the reviewer for bringing up this point. However, we respectfully disagree as we use the word “kinetics” more from an electrochemical viewpoint, which is generally a mass flow throughout a system. (https://en.wikipedia.org/wiki/Electrochemical_kinetics). We describe the transport of dissociated ions, possibly induced by diffusion and migration in the electrolyte, which can result in crystallization. This crystallization changes over time and space, and has its probable origin in the kinetics of the ions.

4) The abstract mentions Li crystallites. Is it this what you mean?

Response: We thank the reviewer for this remark. As we observe Li₂CO₃, LiOH, and lithium crystallites, we generalize the wording to “lithium-based crystallites”.

It reads on page 2:

“In this study, nano-focus X-ray wide-angle scattering (nWAXS) is used to detect possible lithium-based crystallites in the polymer-based electrolyte.”

5) Methods:

a. in the section EC characterization, it is not clear how transference number was measured with EIS, can you clarify.

Response: We thank the reviewer for this remark to help us clarify the determination of the transference number. The transference number was calculated following the Bruce-Vincent standard test (reference #24 in the main text, and see, for example: <https://lithiuminventory.com/experimental-electrochemistry/transference-number-measurement/>). For that, the results of a combined electrochemical impedance curve and a chronoamperometry test are used. We added the formula accordingly.

It reads on page 18:

“The transference number was calculated with the following formula:

$$t = \frac{I_s(\Delta V - I_0 R_0)}{I_0(\Delta V - I_s R_s)}$$

I_s: steady state current, I₀: initial current, ΔV: applied voltage, R₀: resistance before polarization, R_s: resistance after polarization.”

b. FTIR. What sample holder? ATR or transmission? Remember the methods should be described in a way that the experiments can be judged and repeated.

Response: We thank the reviewer for bringing up this point. We measured FTIR in transmission mode. We added a Figure to the Supporting Information and a more detailed description to the Methods part.

It reads on page 19:

“The pristine and lithium-contacted polymers were placed into an airtight FTIR sample holder in an argon-filled glovebox. After that, the respective sample was directly measured at ambient conditions, and the beam was aligned to the respective polymer area. The FTIR sample holder was custom-made from copper to operate FTIR spectroscopy. Since the presented FTIR spectra are performed in transmission mode, the cell is equipped with IR transmissive ZnS windows, which close the sample environment with the help of a sealing O-ring. The copper cell can host sample sizes up to 1 x 1 cm². A sketch and photograph of the cell are given in Supplementary Fig. 5.”

It shows on page 7 of the Supplementary Information:

Supplementary Figure 5. Sketch and photograph of the FTIR cell

Reply to REVIEWERS' COMMENTS NCOMMS-24-74934B

Reviewer #4 (Remarks to the Author):

Comment: The main concern previously arose from the text and Figs. 2-4 which suggested that the beam passed very closely to the Li/polymer interface. Now saying that the beam passed "somewhere" clarified this question.

Also the other comments were well responded to. I'm happy to support publication.

Answer: We thank the reviewer for the time spent and the helpful questions, and we are very delighted that the manuscript is now suggested for publication.